# Light absorption of brown carbon in eastern China based on 3-year multi-wavelength aerosol optical property observations and an improved Absorption Ångström exponent segregation method

Jiaping Wang[1,2,3], Wei Nie[1,2], Yafang Cheng[3,4], Yicheng Shen[1,2], Xuguang Chi[1,2], Jiandong Wang[3], Xin Huang[1,2], Yuning Xie[1,2], Peng Sun[1,2], Zheng Xu[1,2], Ximeng Qi[1,2], Hang Su[3,4,2], and Aijun Ding[1,2*]

[1]Joint International Research Laboratory of Atmospheric and Earth System Sciences, School of Atmospheric Sciences, Nanjing University, Nanjing, 210023, China

[2]Jiangsu Provincial Collaborative Innovation Center for Climate Change, Nanjing, 210023, China

[3]Max-Plank Institute for Chemistry, Germany

[4]Research Institute for Climate and Environment, Jinan University, Guangzhou

*Correspondence to: Aijun Ding (dingaj@nju.edu.cn)

**Abstract**. Brown carbon (BrC), a certain group of organic carbon (OC) with strong absorption from the visible to ultraviolet (UV) wavelengths, makes considerable contribution to light absorption on both global and regional scales. High concentration and proportion of OC has been reported in China, but studies of BrC absorption based on long-term observations are rather limited in this region. In this study, we reported 3-year results of light absorption of BrC based on continuous measurement at the Station for Observing Regional Processes of the Earth System (SORPES) in the Yangtze River Delta, China combined with Mie-theory calculation. Light absorption of BrC was obtained using an improved Absorption Ångström exponent (AAE) segregation method. AAE of non-absorbing coated black carbon (BC) at each time step is calculated based on Mie-theory simulation together with single particle soot photometer (SP2) and Aethalometer observations. By using this improved method, the variation of AAE over time is taken into consideration, making it applicable for long-term analysis. The annual average light absorption coefficient of BrC ($b_{abs\_BrC}$) at 370 nm was 6.3 $Mm^{-1}$ at the SORPES station. The contribution of BrC to total aerosol absorption ($P_{BrC}$) at 370 nm ranged from 10.4% to 23.9% ([10th] and [90th] percentiles, respectively), and reached up to ~33% in open biomass burning-dominant season and winter. Both $b_{abs\_BrC}$ and $P_{BrC}$ exhibited clear seasonal cycles with two peaks in later spring/early

summer (May-June, $b_{abs\_BrC} \sim 6$ Mm$^{-1}$, $P_{BrC} \sim 17\%$) and winter (December, $b_{abs\_BrC} \sim 15$ Mm$^{-1}$, $P_{BrC} \sim 22\%$), respectively. Lagrangian modeling and chemical signature observed at the site suggested that open biomass burning and residential coal/biofuel burning were the dominate sources influencing BrC in the two seasons, respectively.

# 1  Introduction

Atmospheric aerosols not only pose adverse impacts on human health, but also alter earth's radiation balance through their strong light scattering and absorption, substantially influencing regional and even global climate change (IPCC, 2013; Dockery et al., 1993; Wang et al., 2017b). Light absorption of aerosols strongly influence the magnitude and sign of radiative transfer. Black carbon (BC) and dust have been considered as two dominant contributors to aerosol absorption extinction. However, a certain type of organic aerosol defined as brown carbon (BrC) was revealed to be of strong light-absorbing efficiency (Formenti et al., 2003; Pöschl, 2003; Andreae and Gelencser, 2006; Kirchstetter et al., 2004; Mukai and Ambe, 1986; Patterson and McMahon, 1984), which can pose perturbations on radiation transfer similar to BC. This implies that aerosol cooling effect could be overestimated by ignoring the light absorption from BrC. Its strong light absorption in UV range can also affect atmospheric oxidizing capacity by restraining the photolysis rates for photochemically active gases (Laskin et al., 2015). It has been reported that the radiative forcing by BrC globally is about one fourth of that by BC (Feng et al., 2013), while regional radiative forcing by BrC in areas with intensive combustion activities (e.g. South and East Asia, South America, and Africa) can be much higher than global average, indicating the substantial contribution by BrC to aerosol light absorption in these regions and thereby significantly influence regional climate change. Therefore, investigation on BrC contribution to light absorption is of great importance to reduce the uncertainties of aerosol-radiation interaction (ARI) estimation.

BrC is defined as the organic carbon which can absorb solar radiation efficiently in near-UV (300 ~ 400 nm) to visible ranges (Formenti et al., 2003; Pöschl, 2003; Andreae and Gelencser, 2006; Kirchstetter et al., 2004; Mukai and Ambe, 1986; Patterson and McMahon, 1984). It is a group of species with specific physical property but difficult to characterize detailed chemical components. BrC can be produced not only from primary emissions relating to biomass burning and fuel combustion, but also from secondary organic aerosol (SOA) formation through oxidation of volatile organic compounds (VOCs) (Andreae and Gelencsér, 2006; Saleh et al., 2014; Chen and Bond, 2010; Lack et al., 2012; Laskin et al., 2015; Healy et al., 2015). Some studies reported that vehicle and marine emissions may also be the sources of BrC (Stone et al., 2009; Cavalli et al., 2004). The complication

of the emitted mixtures containing BC, BrC, non-absorbing OA, and inorganic materials in different proportions makes it difficult to perform source attribution and to estimate its emission factor.

Light absorption of BrC is usually estimated based on its strong wavelength dependence with higher absorption from visible to UV range. The wavelength dependence of the aerosol absorption coefficient ($b_{abs}$) is normally represented by absorption Ångström exponent (AAE) and its relationship with $b_{abs}$ is $b_{abs}(\lambda) \propto \lambda^{-AAE}$ (Moosmüller et al., 2011;Sun et al., 2007). Based on the difference of wavelength dependence for BC and BrC, previous studies segregated light absorption of BrC from multi-wavelength optical measurements (Lack and Langridge, 2013; Mohr et al., 2013; Shen et al., 2017b), which is called AAE method. Earlier, the similar concept was used to segregate carbonaceous aerosol fractions from different emission sources based on their difference in AAE value (e.g. wood burning and traffic emission) (Sandradewi et al., 2008; Healy et al., 2012). Usually, AAE of BC ($AAE_{BC}$) was set to be 1.0 based on the properties of bulk BC by many previous studies, assuming the unity of $AAE_{BC}$ between any two wavelengths within UV to near inferred (IR) range (Shen et al., 2017b; Olson et al., 2015; Lack and Langridge, 2013). However, the large uncertainty was mentioned by Lack and Langridge (2013) using this assumption and also this method can only be used when the proportion of BrC is high. Moreover, it is noticed that AAE of BC is not always 1.0, instead, it can be affected by particle size and mixing state (Lack and Cappa, 2010; Liu et al., 2015; Liu et al., 2014).

China, the largest country in East Asia with tremendous fossil fuel and biofuel consumption and extensive agricultural burning, is of great concern in terms of its large contribution of carbonaceous aerosols including light absorbing carbon (BC, BrC) (Ding et al., 2016a). BC issue has been noticed in recent years in China, including its temporal variations, emission sources, climate effect (Cao et al., 2006; Cao et al., 2009; Yang et al., 2009; Wang et al., 2017a; Huang et al., 2014) as well as its 'dome effect' in modifying the boundary layer and enhancing haze pollution in megacities (Ding et al., 2016b; Wang et al., 2018). Organic matter(OM) is a large contributor to $PM_{2.5}$ in China (15%-51% of $PM_{2.5}$, Wang et al., 2017c), especially in the Yangtze River Delta (YRD) region, which is one of the most densely populated city cluster in eastern China and also an important agricultural center with crops planted in both cold and warm seasons (Ding et al., 2013c). In YRD region, OM fraction is 20%-40% of $PM_{2.5}$ mass (Wang et al., 2017c; Wang et al., 2017b) due to the influence by complicated combustion

sources, therefore, BrC is a highly possible to be the contributor to aerosol light absorption in this region. However, studies concerning BrC in China, especially YRD region, are still limited up to now. Yuan et al. (2016) reported light absorption contributions of BrC in the Pearl River Delta (PRD) region to be 6.3% to 12.1% at 405nm in autumn and winter campaigns by using the AAE method. Shen et al. (2017b) conducted light absorption measurement in Xi'an in summer and winter, and reported the average $b_{abs\_BrC\_370}$ of 6.4 $Mm^{-1}$ and 43.0 $Mm^{-1}$ in the two seasons, respectively, based on AAE method. Yang et al. (2009) derived mass absorption efficiency (MAE) of BrC at 370 nm which was 2.2 $m^2\ g^{-1}$ and the AAE of BrC was 3.5, respectively during campaign in March 2015 at Xianghe in North China. It is noticed that long-term investigation of BrC light absorption in YRD region has not been reported yet.

This study intends to provide a comprehensive analysis on light-absorbing characteristics of BrC and its influencing factors based on field measurement of multi-wavelength aerosol light absorption at the SORPES station in Nanjing in the western YRD region and an improved AAE segregation method based on Mie-theory simulation and BC single particle measurement. The contribution of BrC to light absorption is estimated based on a 3-year observation and the potential sources of BrC in typical seasons are discussed.

## 2 Methodology

### 2.1 Measurements of optical properties and relevant species

Field observation was conducted at the Station for Observing Regional Processes of the Earth system (SORPES), located in the Xianlin campus of Nanjing University on top of a small hill (118 °57′10″E, 32 °07′14″ N; 40 m a.s.l.) (Shen et al., 2018; Ding et al., 2013c; Ding et al., 2016a). The SORPES station is a research and experiment platform in western YRD, which is under influence of intensive human activities (Ding et al., 2016a). This observation site is about 20 km east of the city center. The map and emission character surrounding the site were given in our previous works (e.g. Ding et al., 2013a; Ding et al., 2016a). Since this observation site is generally upwind from city center and under influence of East Asian monsoon, this site is generally downwind of densely populated city cluster

including the megacity Shanghai. Therefore, this station can be considered as a regional background station in the western YRD region and is an ideal site to study the impact of multiple anthropogenic emission on regional air quality in eastern China (Ding et al., 2016a; Ding et al., 2013a).

Aerosol optical properties were measured from 1 June 2013 to 31 May 2016. Aerosol light absorption measurement was conducted using a multi-wavelength Aethalometer (Model AE-31, Magee Scientific Company Berkeley, California, USA), which performs continuous measurements at seven wavelengths, i.e., 370 nm in ultraviolet (UV) wavelength range (UV), 470 nm, 520 nm, 590 nm, 660 nm, 880 nm in visible (VIS) wavelength range, and 950 nm in infrared (IR) wavelength range. The time resolution is 5 min. Sample air was obtained through a stainless-steel inlet with a $PM_{2.5}$ cut cyclone (Very Sharp Cut Cyclone, VSCC, BGI Inc.) to avoid the impact of coarse mode particles (e.g. dust), protected with a rain cap. The sample flowrate of the aethalometer was set to 5.0 liter per minute (LPM). In this study, the measurements at 370 nm, 520 nm and 880 nm were used for further analysis. The wavelength of 370 nm was used because studies have found that BrC shows strong light absorption near UV wavelength range (Andreae and Gelencsér, 2006; Hansen and Schnell, 2005). Measurement data at the wavelength of 880 nm was chosen because light absorption at this wavelength normally represents the BC absorption (Virkkula et al., 2015). Regarding to the wavelength dependence analysis of BC and BrC, the wavelength near 880 nm is better for the calculation of AAE of BC because BC is the dominant absorption components at that wavelength range. However, the response of the 590 nm and 660 nm data may be affected by the presence of interfering materials such as hematite mineral dust and tobacco smoke (user manual of aethalometer AE-31, Hansen and Schnell, 2005), hence 520 nm data was used for the following calculations. Light scattering coefficients at three wavelengths (450 nm, 525 nm, and 635 nm) were measured by an integrating nephelometer (Aurora 3000, Ecotech). Sample air passed through a stainless tube with a rain cap and an external heater. In order to maintain sample air at low humidity (RH < 50%), an internal heater was used. For those data with RH exceeding 50% due to the malfunction of the heater, scattering coefficients were corrected for hygroscopic growth (Zhang et al., 2015). Light absorption coefficient ($b_{abs}$) at each wavelength $\lambda$ was calculated using the method presented by Collaud Coen et al. (2010) to correct the systematic errors of filter-based absorption measurements. The attenuation coefficient $b_{ATN}$ at each wavelength $\lambda$ is firstly calculated

from

$$b_{ATN}(\lambda) = \frac{A \cdot \Delta ATN(\lambda)}{Q \cdot \Delta t} \qquad Eq.1$$

where A and Q represent the spot size and flow rate, respectively. $\Delta ATN(\lambda)$ is the attenuation change in time step $\Delta t$. $b_{abs}$ at wavelength $\lambda$ is then obtained after correction for filter-loading effect, embedded aerosol scattering effect and multiple scattering effect by the filter fiber. The correction is performed using the Collaud Coen correction algorithm with Schmid scattering correction adopted (Schmid et al., 2006; Collaud Coen et al., 2010). The equation can be presented as

$$b_{abs}(\lambda) = \frac{b_{ATN}}{(C_{ref} + C_{scat}) \cdot R} \qquad Eq. 2$$

where R is the function for filter-loading correction calculated using the equation from Collaud Coen et al. (2010) (Eq. 13). $C_{scat}$ represents the aerosol scattering correction. To calculate $C_{scat}$, light scattering coefficients and scattering Ångström exponents measured by nephelometer are used to obtain scattering at the aethalometer wavelengths, and the constants to calculate $C_{scat}$ are taken from Arnott et al. (2005). Detailed calculation equations of R and $C_{scat}$ can be found in Collaud Coen et al. (2010). $C_{ref}$ is the multiple scattering correction factor, which is set to be 4.26 according to Collaud Coen et al. (2010). A comparison of different Aethalometer correction algorithms (Saturno et al., 2017) shows that AAE derived by Collaud Coen correction algorithm agrees well with that from multi-wavelength reference measurement, proving the reliable AAE values from this correction. Collaud Coen correction also shows a good performance in obtaining absorption coefficients at 370 nm (Saturno et al., 2017), which is the critical wavelength in BrC segregation. Absorption coefficients are presented under Standard Temperature and Pressure (STP, i.e. 273.15 K, 1013 hPa).

Size distribution and mixing states of refractory BC was measured using the single particle soot photometer (SP2, Droplet Measurement Technologies, USA). The operation mechanism of SP2 has been well described in previous studies (Stephens et al., 2003; Schwarz et al., 2006, 2008). SP2 uses a laser induced incandescence technique which equips with a Nd:YAG laser ($\lambda = 1064$ nm) and optical detectors to quantify the size of single particle by detecting scattering and laser induced incandescence signal. The mass of refractory BC can be determined by its nearly linear relationship with the peak

height of incandescence signal. The incandescence response of SP2 was calibrated using fullerene soot with known mobility diameter selected by DMA (Gysel et al., 2011). The mass of fullerene soot was calculated using size-resolved effective density presented by Gysel et al. (2011). For pure scattering particles, the peak height of the scattering signal is linearly proportional to particle scattering cross-section. As for BC-containing particles, due to the loss of coating and the vaporization of BC core when the particle passes through the laser beam, the scatting signal is different from the original particle. To determine the scattering cross-section of BC-containing particles and saturated scattering particles, the leading-edge-only (LEO) fit method developed by Gao et al. (2007) was adopted. $D_p$ of BC was then estimated by using core-shell Mie model assuming BC core and shell refractive index of 2.26-1.26i (Moteki et al., 2010) and 1.52-0i (Pitchford et al., 2007), respectively. The scattering signal was calibrated using polystyrene latex spheres (PSL) with known sizes. The detection range of BC core is 80~600 nm assuming a density of 1.8 g cm$^{-3}$ (Bond and Bergstrom, 2006). In this study, the four measurement periods are 20 May~12 June 2016, 8~31 August 2017, 1~30 November 2017 and 1~28 February 2018, representing spring, summer, autumn and winter, respectively.

In present study, water-soluble ions ($K^+$, $Cl^-$) were measured by the Monitor for Aerosols and Gases in Air (MARGA, Metrohm Co.), PM$_{2.5}$ and meteorological data were used for further supporting discussions. More detailed descriptions of these measurements can be found in Ding et al. (2013b, c; 2016a).

**2.2 Optical calculation**

It has been proved that BrC shows strong light absorbance in UV-visible wavelength range. To quantify the light absorption of BrC based on optical measurement results, the AAE segregation method is used (Lack and Langridge, 2013; Mohr et al., 2013). Light absorption of BrC is calculated as the result of $b_{abs}$ minus light absorption coefficient of BC ($b_{abs\_BC}$) at 370 nm. Here $b_{abs\_BC\_370}$ is defined as the absorption coefficient of pure BC or BC with non-absorbing coating at 370 nm, and $b_{abs\_BrC}$ is obtained from total absorption at 370 nm deducting absorption of BC core and lensing effects, as following equation shows

$$b_{abs\_BC\_370} = b_{abs\_880} \times (880/520)^{AAE_{BC520-880}} \times (520/370)^{AAE_{BC370-520}} \qquad \textit{Eq. 3}$$

$$b_{abs\_BrC} = b_{abs\_370} - b_{abs\_BC\_370} \qquad \textit{Eq. 4}$$

where $b_{abs\_370}$ and $b_{abs\_880}$ represents the absorption coefficients at 370 nm and 880 nm, respectively, which is calculated from the light absorption measurement data. $AAE_{BC520-880}$ and $AAE_{BC370-520}$ stands for the AAE of pure BC and BC with non-absorbing coating at long and short wavelength ranges. We calculate $AAE_{BC520-880}$ and $AAE_{BC370-520}$ from SP2 data using core-shell Mie model, which has been widely applied in BC related studies (Bond and Bergstrom, 2006). It is mentioned that BC morphology can affect AAE (Liu et al., 2015) and it is possible to overestimate BrC absorption, however, the complex morphology can vary with time and currently it is hard to evaluate its quantitative effect. Also, this site is a regional background station influenced more by aged air plumes (Ding et al., 2016a), therefore here we implement core-shell model. AAE at two wavelengths is calculated as the following equation:

$$AAE_{\lambda1-\lambda2} = -\frac{\ln(b_{abs_{\lambda1}}) - \ln(b_{abs_{\lambda2}})}{\ln(\lambda1) - \ln(\lambda2)} \qquad \textit{Eq. 5}$$

As mentioned above, AAE value of 1.0 was adopted for BC by many researches. However, $AAE_{BC}$ can vary with BC core size, coating thickness, morphology, etc. Evidences showed that AAE of pure BC cores can be lower than 1.0 as the diameter is out of the range of Rayleigh theory, and that BC with clear shell can possibly have AAE higher than 1.0 (Bond et al., 2013; Lack and Cappa, 2010; Gyawali et al., 2009). It is also observed at the SORPES station that $AAE_{520-880}$, which is expected to be mainly affected by BC absorption, is not always 1.0 and exhibits clear seasonal and diurnal variations (Shen et al., 2018). Hence, assuming $AAE_{BC}$ of 1.0 in the estimation of BrC may induce large uncertainties or bias (comparison of calculated $b_{abs\_BrC}$ assuming $AAE_{BC} = 1.0$ versus the modified method will be discussed later). Therefore, it is essential to firstly evaluate the quantitative impacts of BC size and coating on AAE value and determine the proper $AAE_{BC}$ for more accurate $b_{abs\_BrC}$ calculation.

# 3 Estimation of BC optical properties and BrC segregation

Based on core-shell Mie-theory model, we conducted a series of calculations to discuss the variation pattern of AAE for BC-containing particles (Bohren and Huffman, 1983). We used Christian Mätzler's code (Christian Mätzler, 2002) for Mie calculations of spherical particles at different wavelengths.

Mie-theory simulations were conducted firstly for mono-dispersed particles. Here particle core diameter ($D_c$) is defined as the diameter of the core alone and the shell diameter ($D_p$) refers to the total particle diameter. Coating thickness was represented using $D_p/D_c$. $D_c$ increases from 1 to 200 nm with 1 nm interval and $D_p/D_c$ was set to be 1.0~3.0 with 100 bins. The refractive index (RI) of BC core was

10 set to be 1.56-0.47i according to Dalzell and Sarofim (1969), and it was 1.52-0i for clear shell (Pitchford et al., 2007).

Figure 1 shows the variations of $AAE_{BC370-520}$ and $AAE_{BC520-880}$ with $D_c$ and $D_p/D_c$ of mono-dispersed BC-containing particles. The black dash lines in the figure illustrate the $D_c$ and $D_p/D_c$ value range where BC mostly distributed at the SORPES station (shown in Figure 2). Within this range, it

can be found that $AAE_{BC370-520}$ and $AAE_{BC520-880}$ mainly exhibit a deceasing pattern as $D_c$ increases but with different amplitude. Taking pure BC as an example, $AAE_{BC370-520}$ decreases from 1.1 to 0.6 as $D_c$ increases from 80 nm to 180 nm, while $AAE_{BC520-880}$ decreases from 1.2 to 1.0 as $D_c$ increases. While as $D_p/D_c$ grows, AAE of BC shows a non-monotonously variation trend. For $D_c$ = 100 nm, as $D_p/D_c$ grows from 1 to 3, $AAE_{BC370-520}$ first increases from 1.0 to 1.3, peaking at $D_p/D_c$ = 2 and then decreases

back to 1.0 at $D_p/D_c$ = 3. While $AAE_{BC520-880}$ increases with $D_p/D_c$ at Dc=100 nm. As for the magnitude of AAE, $AAE_{BC370-520}$ is generally lower than $AAE_{BC520-880}$ for BC with same size and coating thickness. Above results suggest that the $D_c$ and $D_p/D_c$ range of BC measured in this study is located in the regime where $AAE_{BC}$ changes largely and non-monotonously. Moreover, in short and long wavelength ranges, $AAE_{BC}$ with same BC size and coating thickness is also different. Therefore, instead of assuming AAE

as a constant, real-time $AAE_{BC}$ determination is proposed in this study.

To explore the characteristics of $AAE_{BC}$ at short and long wavelength ranges at this observation site, we analyzed the size distribution and mixing state of BC measured by SP2 firstly. Fig. 2 shows the overall $D_c$ number size distribution and $D_p/D_c$ of BC-containing particles in four seasons. It can be

found that the number size distribution of BC cores in spring, summer and autumn are in similar pattern. In winter, larger BC cores take up a higher proportion than other seasons. However, the coating thickness of BC is relatively lower in winter (peak number at $D_p/D_c \sim 1.6$), possibly due to low photochemical oxidation in this season. The coating thickness of BC is higher in spring than other seasons.

Based on the size and coating thickness of each BC-containing particle measured by SP2, $b_{abs}$ was calculated using core-shell Mie model. $AAE_{BC370-520}$ and $AAE_{BC520-880}$ were then derived as Figure 3a illustrates. It can be observed that the fluctuation of $AAE_{BC}$ at both short and long wavelength range in different seasons is not significant. Median values of $AAE_{BC520-880}$ in spring, summer, autumn and winter are 0.80, 0.78, 0.79 and 0.81, respectively, and $AAE_{BC370-520}$ median values are 0.53, 0.54, 0.51 and 0.50 in four seasons. Moreover, for BC at this site, $AAE_{BC370-520}$ is always lower than $AAE_{BC520-880}$. Figure 3b shows the calculated AAE of particles measured by the Aethalometer. The median values of $AAE_{520-880}$ in four seasons range from 0.83 to 1.03, while $AAE_{370-520}$ shows a higher level and larger variation. The seasonal median values range from 1.00 to 1.31. Compared to Fig. 3a, it is noticed that observed $AAE_{520-880}$ is comparable to $AAE_{BC520-880}$, indicating that BC is the dominant absorbing component in this wavelength range. Contrarily, the difference between $AAE_{370-520}$ and $AAE_{BC370-520}$ is more obvious. Unlike $AAE_{BC}$, $AAE_{370-520}$ is higher than $AAE_{520-880}$ in all seasons, indicating the possibility of presence of other particle component with strong light absorption at short wavelength range and high AAE.

Since normally the SP2 is not used as a long-term continuous observation instrument, an alternative method to derive $AAE_{BC}$ is needed, especially for tracing back the historical level of BrC absorption without real-time SP2 measurement. As shown in Fig. 3, the difference between $AAE_{BC370-520}$ and $AAE_{BC520-880}$ at this site is not significant over time. Hence a correction factor $R_{AAE}$ is defined as the ratio of $AAE_{BC370-520}$ to $AAE_{BC520-880}$ calculated from SP2 data. Figure S2a (in Supplementary Information, SI) illustrates the variation of $R_{AAE}$. During the whole observation period, $R_{AAE}$ ranges between 0.60-0.69 (5[th] and 95[th] percentile), and the median value is 0.66, 0.69, 0.64 and 0.62 in spring, summer, autumn and winter, respectively. As mentioned above, $AAE_{520-880}$ calculated from Aethalometer data is approximately equal to $AAE_{BC520-880}$ in the wavelength range where main

absorber is BC (Lack and Langridge, 2013). Therefore, $AAE_{520-880}$ is used to represent $AAE_{BC520-880}$ and real time $AAE_{BC370-520}$ can be derived as *Eq. 6*.

$$AAE_{BC370-520\_real-time} = AAE_{520-880} \times R_{AAE} \qquad\qquad Eq.\ 6$$

Then, $b_{abs\_BrC}$ can be derived from *Eq. 4* and *Eq. 6*. Based on field observations at this site, we set $R_{AAE}$ as 0.65, which is the mean value of the whole time for the following calculation. To examine the sensitivity of $b_{abs\_BrC}$ to $R_{AAE}$ value, we calculated $b_{abs\_BrC}$ using different $R_{AAE}$ value. $R_{AAE}$ was set to be 0.1~1.0 with 0.05 interval and the overall average $b_{abs\_BrC}$ is plotted in Figure S2b. The dash lines are the lower and upper limit of $R_{AAE}$ (0.60 and 0.69, the 5[th] and 95[th] percentile) from SP2 measurement and the corresponding $b_{abs\_BrC}$. Therefore, the shaded area represents the uncertainty range due to the different $R_{AAE}$, which is approximately $\pm$ 7%. Time series of $b_{abs\_BrC}$ is shown in Figure 4. Calculated $b_{abs\_BrC}$ using $AAE_{BC} = 1.0$ is also plotted in the figure as the black solid line. Also, as shown in Fig. 4, calculating $b_{abs\_BrC}$ assuming $AAE_{BC}=1$ leads to a large amount of negative values, especially when light absorption of BrC is low. While by using modified method, long-term $b_{abs\_BrC}$ can be obtained with satisfactory data validity.

## 4   Long-term characteristics of BrC light absorption at the SORPES station

Based on above results, $b_{abs\_BrC}$ was determined with one-hour time interval following *Eq. 3* and *Eq. 4*. $P_{BrC}$, defined as the contribution of light absorption by BrC at 370 nm ($P_{BrC} = \frac{b_{abs\_BrC}}{b_{abs\_370}}$), was also calculated using the measurement data. Statistical overview is summarized in Table 1. Seasonal cycles of $b_{abs\_BrC}$ and $P_{BrC}$ are shown in Figure 5, together with, $K^+$, $K^+/PM_{2.5}$, $PM_{2.5}$ and AAE at different wavelength ranges. Firstly, it can be found that $b_{abs\_BrC}$ exhibits a distinct two-peak seasonal pattern where the peak value occurs in June and December, with mean $b_{abs\_BrC}$ of 5.9 Mm$^{-1}$ and 15.5 Mm$^{-1}$, respectively (Fig. 5a). It is also observed that $b_{abs\_BrC}$ during winter, especially December, is much higher than that in other three seasons (two to three times higher). $P_{BrC}$ also presents a two-peak seasonal trend with the high $P_{BrC}$ months of May-June and December. The mean $P_{BrC}$ in winter and

summer are 19.6% and 14.4%, respectively, which is lower than that in Xi'an but higher than the PRD region (Shen et al., 2017b;Yuan et al., 2016). The $95^{th}$ percentile of $P_{BrC}$ can reach to 32% in December, which certainly cannot be ignored in light absorption estimation in the YRD region. Notably, $P_{BrC}$ has a similar seasonal variation pattern with $K^+$, except in February when intensive fireworks during the Chinese New Year can lead to significantly high values of $K^+$ concentrations. Moreover, $AAE_{370-520}$ shows distinct seasonal variations, which has much wider range of changing with $1^{th}$ and $99^{th}$ percentile of 0.6 and 1.9 (Table 1), than that of $AAE_{520-880}$. Also noticed is that the variation pattern of $AAE_{370-520}$ is similar with $K^+$. Since $K^+$ is mainly emitted from primary combustion processes, the simultaneous variation of $P_{BrC}$ and $AAE_{370-520}$ with $K^+$ suggests that primary emissions are likely to make considerable contribution to BrC in this area for the most of time.

Due to the distinct seasonal trend of $b_{abs\_BrC}$ observed at the SORPES station and its considerable contribution to the light absorption, it is essential to recognize the potential source areas and types of BrC in the YRD region. The diurnal variation of $b_{abs\_BrC}$ was compared to $b_{abs\_BC}$ in each season, shown in Figure 6. Hourly mean values of $b_{abs\_BrC}$ and $b_{abs\_BC}$ are plotted. The reason to compare $b_{abs\_BrC}$ with $b_{abs\_BC}$ is that BC is one of the major light absorbers in the atmosphere and it is mostly from primary emission sources. Overall, $b_{abs\_BrC}$ shows quite similar diurnal pattern with $b_{abs\_BC}$ in four seasons, which is high during night and start decreasing after sunrise. This indicates that BrC at the site is also dominated by primary emissions. The lowest values occurred in the afternoon due to the development of Planetary Boundary Layer (PBL). It is observed that $b_{abs\_BC}$ exhibits clear morning peaks in all four seasons, suggesting that traffic exhaust is likely to be a considerable emission source of BC at this site. Compared to that, the morning peaks of $b_{abs\_BrC}$ in summer and autumn is less obvious than that of BC, while in winter and spring the morning peaks are not noticeable. Such difference reveals that during winter and spring, traffic emission is not the main contributor to local BrC.

In order to investigate the potential source region of BrC at the SORPES station, Lagrangian particle dispersion modeling (LPDM) were conducted following the method developed by Ding et al. (2013b) by using the Hybrid Single-Particle Lagrangian Integrated Trajectory (HYSPLIT) model (R Draxler and Hess, 1998;Stein et al., 2015). The meteorological data used in this model was GDAS (Global Data Assimilation System) data with a spatial resolution of 0.05° in both latitude and longitude.

In each simulation, 3000 particles were released at an altitude of 100 m above the ground level (Wang et al., 2017a) and backwardly run for a 3-day period, and then the retroplume, i.e. footprint of surface 100 m, were obtained following the method of Ding et al. (2013b).

As shown in Fig. 5a, the two distinct peaks of $b_{abs\_BrC}$ is in June and December, respectively. It is then necessary to explore the possible emission sources in these two months. As mentioned before, since BrC is an operational definition, it is difficult to perform source apportionment for BrC. The feasible way to determine its sources is to compare the relationships between BrC and certain species that are possibly from the same emission sources. Since the emission of BrC is usually related to biomass burning (Laskin et al., 2015;Saleh et al., 2014), maps of fire counts in June and December of 2014 are presented in Figure 7, together with monthly averaged 3-day backward retroplume in order to firstly diagnose whether the majority of air plumes pass through open burning areas in these two months. Then, correlations between $b_{abs\_BrC}$ and $K^+$ in June and December were compared since $K^+$ is normally considered as a tracer of primary emission from BB (Ding et al., 2013a). The results are shown in Figure 7 and Figure 8.

Fig. 7a shows intensive open burning, detected in June from the northwest of the site, but in this month air plumes are mainly transported from the eastern area of Nanjing where fire spots can also be found but less concentrated than northwestern regions. Contrarily, very few fire counts can be detected in December, suggesting the much less open burning events in this month. Therefore, the high level of $b_{abs\_BrC}$ in December is not likely to be from open BB emissions. Map of retroplume reveals that air masses are mainly from the north area in December. Then, correlations between daily average $b_{abs\_BrC}$ and $K^+$ mass concentration in June and December is compared as Fig. 8a shows. It can be observed that the correlations between $b_{abs\_BrC}$ and $K^+$ in these two months exhibit a clearly different pattern. The slope of fitted $b_{abs\_BrC}$ and $K^+$ in June is 4.65 and the correlation coefficient $R^2$ is 0.92. Knowing the strong correlation between $b_{abs\_BrC}$ and $K^+$ in June, combined with the observed intense fire counts in this month, it can be presumed that primary open BB emission can be a major contributor to BrC during June. As for December, $b_{abs\_BrC}$ and $K^+$ presents a much higher slope (slope is 10.59), which is approximately twice as that in June. The distribution of $b_{abs\_BrC}/K^+$ in these two seasons are shown in

Fig. 8b. The result displays that $b_{abs\_BrC}/K^+$ of June and December have a significant difference (through t-test, $P < 0.05$), indicating that the dominant emission source of BrC in December is not open biomass burning (significant difference test of $b_{abs\_BrC}/K^+$ is also done for May and June, which are the main open BB seasons for comparison, and the result shows that there is no significant difference in these two months, Fig. S5). Above analyses have demonstrated that BrC is not mainly from open biomass burning and vehicle exhausts in December. The increase of $P_{BrC}$ in winter, representing the higher ratio between BrC to BC mass, indicates the change of main emission sources. Nanjing is dominated by northeasterly wind in winter (Figure S6) with air masses long-range transported from North China by winter monsoon (Ding et al., 2013c, 2016a). Because of cold weather in winter, there is higher residential coal and biomass/biofuel burning emission for household heating (Fu et al., 2018). Therefore, residential coal and household biomass burning can possibly be the main sources of high BrC in December under the influence of winter monsoon (Zhang et al., 2016). Studies conducted in Beijing have also suggested the important contribution of residential biofuel (Yan et al., 2015; Cheng et al., 2016) and coal combustion (Yan et al., 2017) on BrC in northern China in winter. From above analyses, it can be concluded that the high BrC absorption contributions in May-June and winter season are caused by different sources. In May-June, strong open biomass burning leads to the high BrC in Nanjing, while in winter, open biomass burning and vehicle emissions make small contribution to the high BrC at observation site. Residential coal and household biofuel burning are possibly the major sources of BrC in winter season. Detailed sources of BrC can be further explored combining field measurement of organic aerosols in the future.

## 5  Summary

In this study, light absorption of BrC was quantified using the optical method based on the definition of BrC. Mie-theory simulation and observational results were combined to improve this method by calculating $AAE_{BC}$ at each time point instead of assuming a constant. Long-term variation of $b_{abs\_BrC}$ and $P_{BrC}$ were then derived. Apparent light absorption contributed by BrC is discovered in YRD region. $b_{abs\_BrC}$ and $P_{BrC}$ both exhibit clear seasonal cycles with two peaks in May to June and December. The light absorption contribution of BrC at 370 nm ranges from 10.4% to 23.9% (for 10th and 90th

percentiles), and can reach to 33.3% in open BB-dominant season and winter season. Comparison between $b_{abs\_BrC}$ and $b_{abs\_BC}$ suggests that vehicle emission makes negligible impact on regional BrC level during winter and spring. Source analysis was performed based on temporal variations of BrC and the comparison of possible co-emitted pollutants or related parameters. Lagrangian particle

dispersion modeling (LPDM) and MODIS fire data were also used to support analysis. The month of June and December with the peak level of $b_{abs\_BrC}$ are chosen to analyze the potential emission sources of BrC and it is found that the high contributions of BrC in these two months are dominated by different emission sources. In June, intensive primary open BB emission is the dominant source of BrC, making $b_{abs\_BrC}$ appears short-time high values raising its average level in this month. While in December, high

portion of BrC is possibly contributed by residential coal and household biofuel burning.

Overall, this work explores an improved optical method to quantify $b_{abs\_BrC}$ from long-term observation. Compared to the conventional one, which may induce large uncertainty due to the assumption of constant $AAE_{BC}$ regardless of its variation with particle size, wavelength and time, this improved method is applicable for those sites where BrC proportion is low and make it available for

long-term analysis. This study also highlights the considerable contribution of BrC to light absorption at near UV range in the YRD region. Moreover, different emission sources of BrC is found in different seasons, providing a clearer reference for mitigation measures as well as regional control policies in eastern China.

**Acknowledgements:**

This work was supported by National Key Research & Development Program of China supported by Ministry of Science and Technology of the People's Republic of China (2016YFC0200500; 2016YFC0202000), the National Natural Science Foundation of China (91544231, 41725020, and 41422504), and the Public Welfare Projects for Environmental Protection (201509004).

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

**Figure captions**

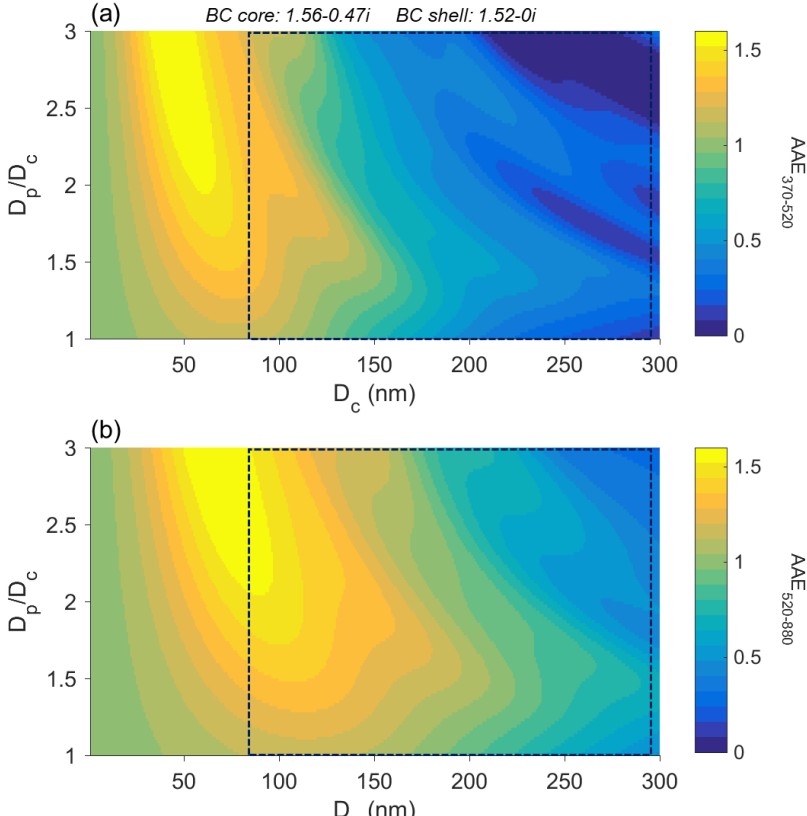

**Figure 1.** Variation of Absorption Ångstrom Exponents between **(a)** 370 and 520 nm (AAE$_{BC370-520}$) and between **(b)** 520 and 880 nm (AAE$_{BC520-880}$) along with black carbon (BC) core diameter (D$_c$) and the coating thickness (D$_p$/D$_c$) of clear (pure scattering) shell simulated with core-shell Mie model. The refractive index (RI) of BC core is set to be 1.56-0.47i according to Dalzell and Sarofim (1969) and RI is 1.52-0i for clear shell (Pitchford et al., 2007).

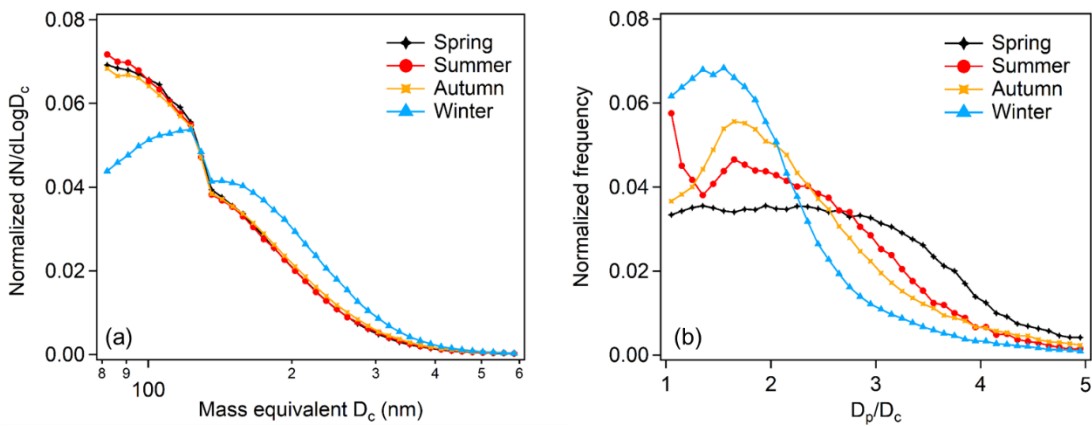

**Figure 2. (a)** Normalized number size distributions of BC core ($D_c$) and **(b)** $D_p/D_c$ of BC-containing particles from

SP2 measurement in four seasons

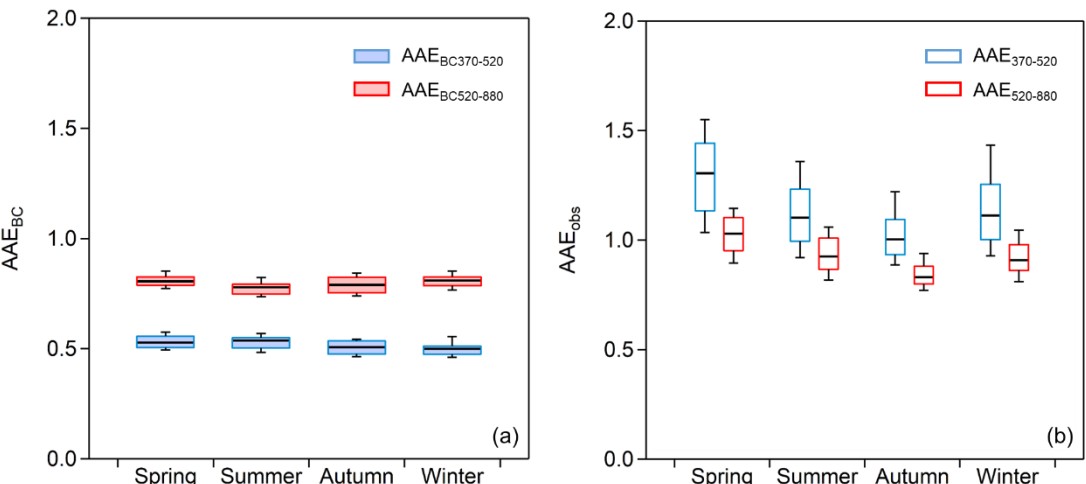

**Figure 3. (a)** $AAE_{BC370\text{-}520}$ and $AAE_{BC520\text{-}880}$ in different seasons calculated using BC size distribution from SP2 and core-shell Mie model; **(b)** $AAE_{370\text{-}520}$ and $AAE_{520\text{-}880}$ from observation using Aethalometer

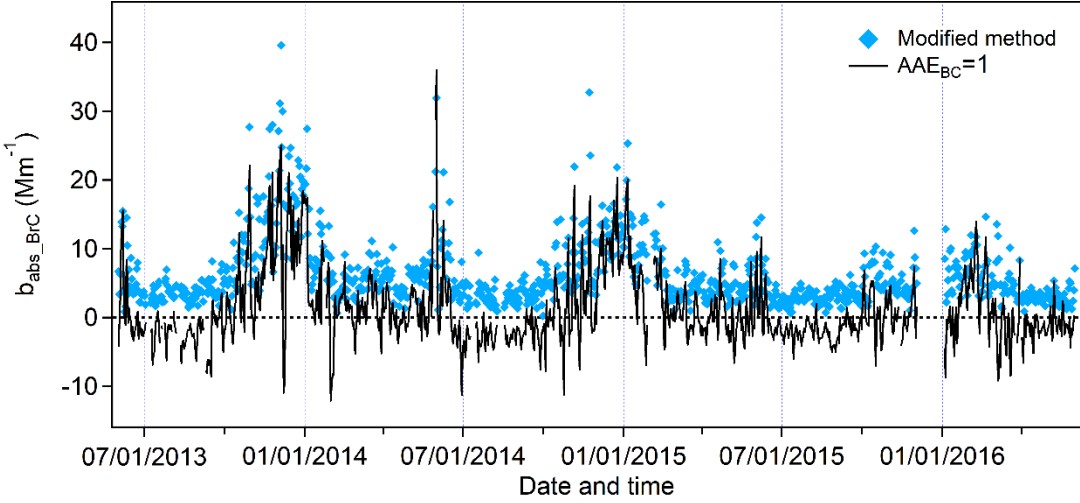

**Figure 4.** Comparison between the time series of daily mean $b_{abs\_BrC}$ using the modified method and using $AAE_{BC} = 1.0$. The blue diamonds represent the calculation result using $R_{AAE} = 0.65$, which is the mean value from SP2 data. Calculated $b_{abs\_BrC}$ using $AAE_{BC} = 1.0$ is plotted as the black line

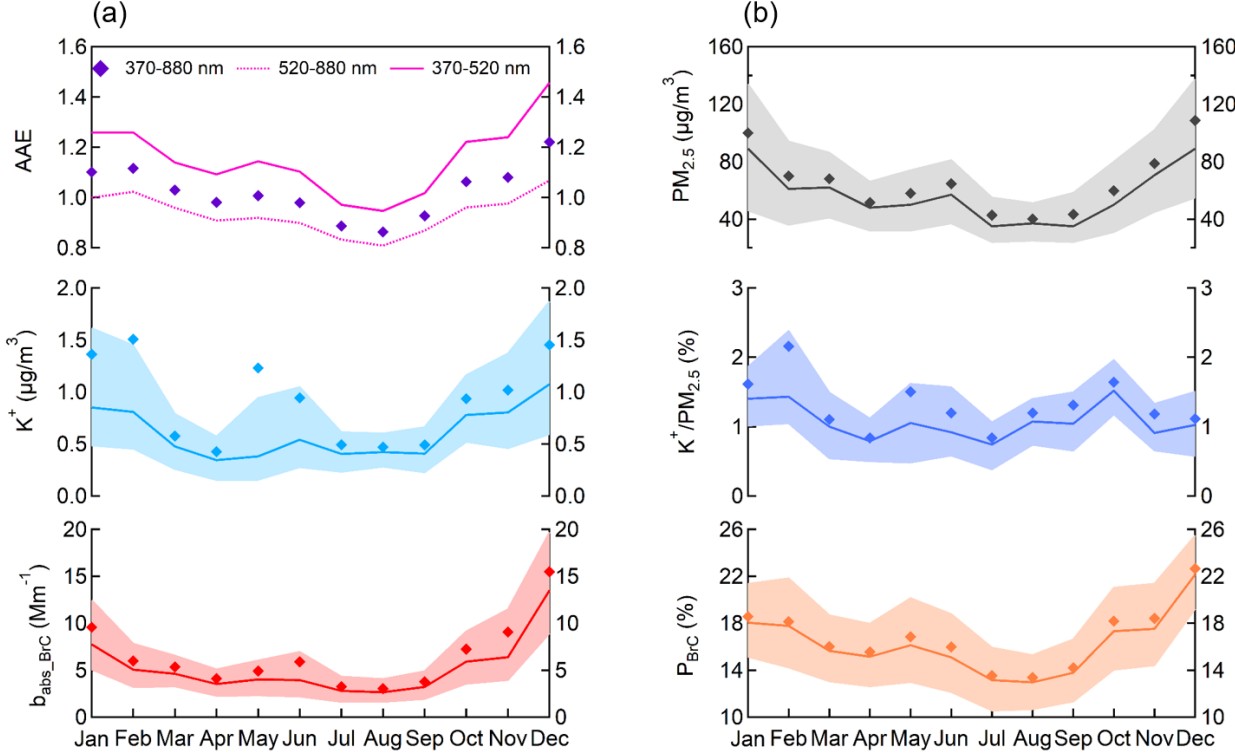

**Figure 5.** Seasonal cycle of **(a)** $b_{abs\_BrC}$, $K^+$, and AAE at different wavelength ranges (AAE$_{370-520}$, AAE$_{370-880}$ and AAE$_{370-880}$, shown as solid line, dash line and diamonds, respectively) and **(b)** $P_{BrC}$, $K^+/PM_{2.5}$ and $PM_{2.5}$. For $b_{abs\_BrC}$, $K^+$, $P_{BrC}$, $K^+/PM_{2.5}$ and $PM_{2.5}$ figures, bold solid lines represent median values, diamonds show the monthly averages and thin solid lines forming the shaded area are 25$^{th}$ and 75$^{th}$ percentiles

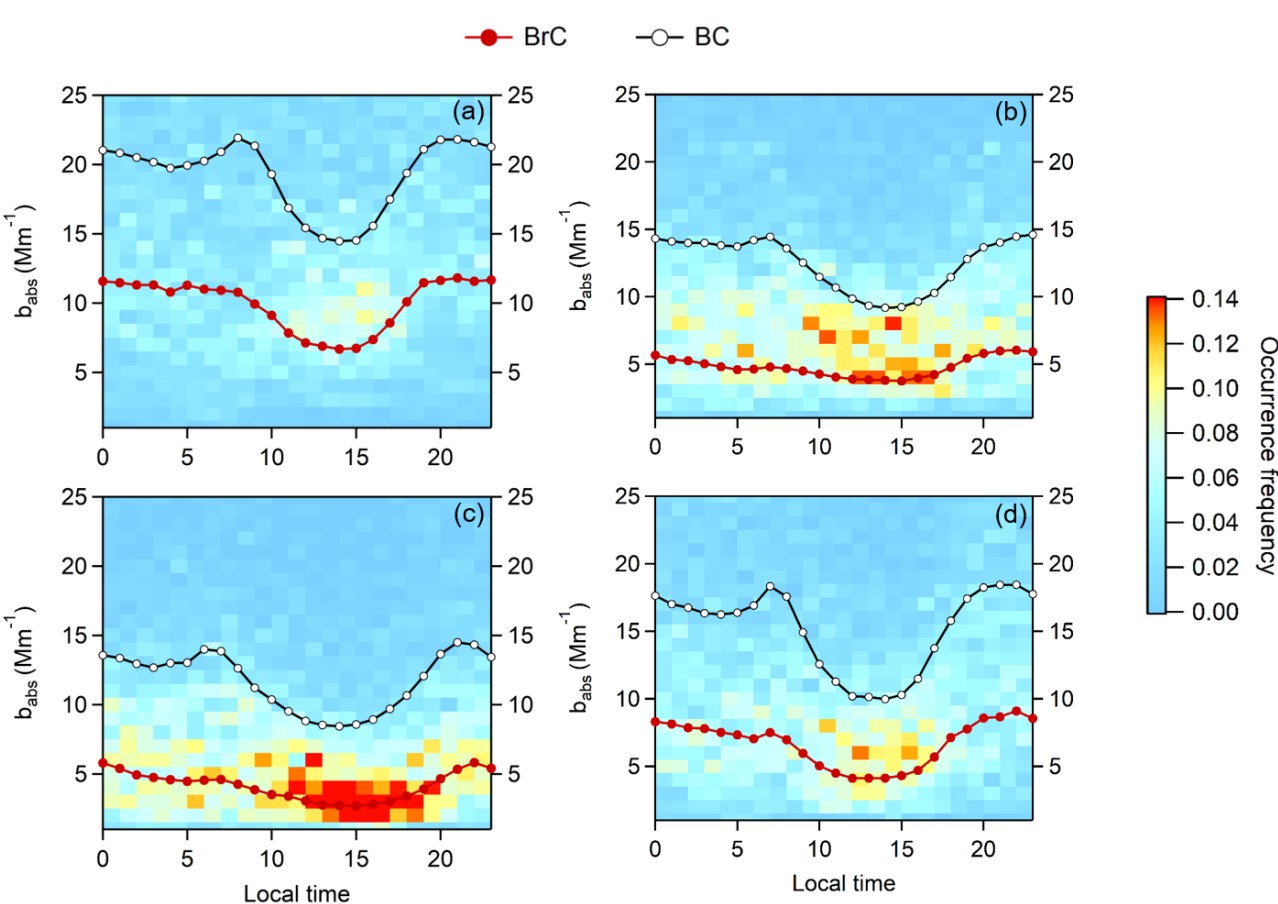

**Figure 6.** Diurnal variations of $b_{abs\_BrC}$ and $b_{abs\_BC}$ in four seasons **(a)** winter, **(b)** spring, **(c)** summer, and **(d)** autumn, respectively. The image plot shows the occurrence frequencies of $b_{abs\_BrC}$ in each $b_{abs\_BrC}$ bins. The dark red and black circle lines represent the hourly mean $b_{abs\_BrC}$ and $b_{abs\_BC}$, respectively.

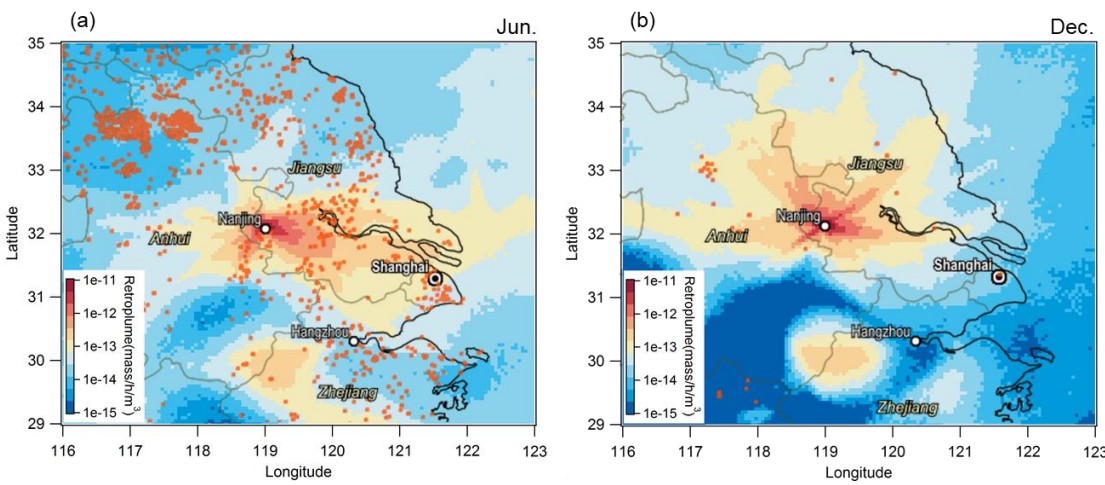

**Figure 7.** Map of averaged 3-day backward retroplume and the fire counts for **(a)** June and **(b)** December in 2014.

Fire count data is from MODIS Collection 6 Active Fire Product provided by NASA fire mapper, downloaded in

2017 (https://firms.modaps.eosdis.nasa.gov/firemap/)

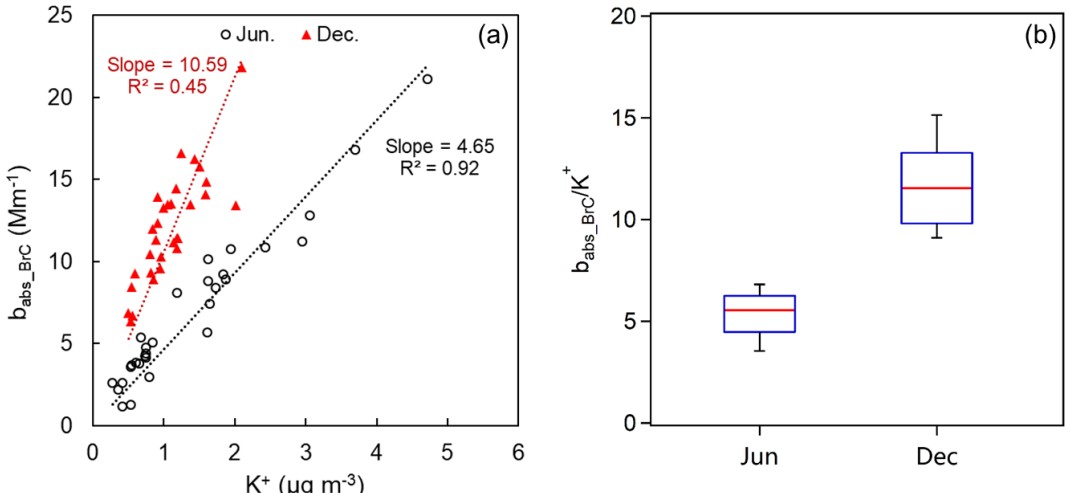

**Figure 8. (a)** Correlations between daily average $b_{abs\_BrC}$ and $K^+$ mass concentration in June (black circles) and December (red triangles); **(b)** Boxplot of $b_{abs\_BrC}/K^+$ in June and December (data is from the year 2014), where red lines represent the median value, blue boxes represent 25th and 75th percentile ranges and thin bars are 5th and 95th percentiles

**Table 1.** Statistical summary of data measured at SORPES station

| | Mean | percentiles | | Seasonal mean | | | |
|---|---|---|---|---|---|---|---|
| | | 1th | 99th | winter | spring | summer | autumn |
| $b_{abs\_BrC}$ (Mm$^{-1}$) | 6.3 | 0.6 | 29.7 | 9.9 | 4.8 | 4.1 | 6.7 |
| $P_{BrC}$ (%) | 16.7 | 6.3 | 33.3 | 19.6 | 16.1 | 14.4 | 17.0 |
| $b_{abs\_BC}$ (Mm$^{-1}$) | 14.5 | 2.4 | 51.6 | 19.2 | 12.5 | 11.7 | 15.1 |
| $b_{abs\_370}$ | 35.8 | 5.6 | 136.0 | 51.0 | 29.7 | 26.5 | 37.3 |
| $b_{abs\_520}$ | 23.9 | 3.9 | 86.3 | 32.8 | 20.3 | 18.5 | 24.8 |
| $AAE_{370-520}$ | 1.2 | 0.6 | 1.9 | 1.3 | 1.1 | 1.0 | 1.2 |
| $AAE_{520-880}$ | 0.9 | 0.6 | 1.2 | 1.0 | 0.9 | 0.8 | 0.9 |
| $K^+$ (µg m$^{-3}$) | 0.9 | 0.1 | 6.5 | 1.4 | 0.7 | 0.6 | 0.8 |
| $Cl^-$ (µg m$^{-3}$) | 2.2 | 0.0 | 12.9 | 4.0 | 2.0 | 0.8 | 1.7 |

## Supplementary Information

To explain the varying AAE of pure BC particles, optical interpretation is performed based on Mie-theory as shown in , where the wavelengths ($\lambda_1$ and $\lambda_2$) 370 nm and 520 nm are used as an example. Firstly, for a given two wavelengths $\lambda_1$ and $\lambda_2$, $AAE_{\lambda1-\lambda2}$ can be calculated from *Eq. 5*, where $b_{abs} =$ MAE $\cdot \frac{\pi\rho}{6} \cdot D_c{}^3$. Therefore, *Eq. 5* can be transferred into the following equation:

$$\text{AAE}_{\lambda1-\lambda2} = - \frac{\ln(MAE_{\lambda1} \cdot \frac{\pi\rho}{6} \cdot D_c{}^3) - \ln(MAE_{\lambda2} \cdot \frac{\pi\rho}{6} \cdot D_c{}^3)}{\ln(\lambda1) - \ln(\lambda2)} = - \frac{\ln(MAE_{\lambda1}) - \ln(MAE_{\lambda2})}{\ln(\lambda1) - \ln(\lambda2)} \qquad Eq.\ S1$$

that is, $\text{AAE}_{\lambda1-\lambda2} \propto \Delta \ \ln(MAE)_{\lambda1-\lambda2}$, as shown in    where MAE is plotted in logarithmic axis. When Dc << λ, the entire particle mass participates in absorption and MAE is a constant, while for Dc >> λ, only the particle's skin contributes to absorption and MAE is inversely proportional to Dc (Bond and Bergstrom, 2006; Moosmuller and Arnott, 2009), therefore, the overall changing pattern of MAE is firstly keeping steady and then drop as a function of Dc. The slight peak of MAE before dropping is due to internal resonances (Moosmüller et al., 2009). Hence, whether AAE increases or decreases with Dc can be determined by comparing the first derivative of MAE at $\lambda_1$ and $\lambda_2$ (shown in the lower axis in Figure S1), which represents the slope of MAE for each Dc. The crossing point of slope_MAE is therefore corresponding to the maximum $\text{AAE}_{\lambda1-\lambda2}$, with core size of $Dc_{max}$. For example, when $\lambda_1$ and $\lambda_2$ are 370 nm and 520 nm, the maximum $AAE_{370-520}$ occurs when $Dc_{max} = Dc_0 = 75$ nm. AAE increases with Dc when $Dc < Dc_0$ but decreases when $Dc > Dc_0$. Since the slope_MAE at different wavelengths are in the same shape only shifting horizontally with longer wavelength, for AAE between longer wavelengths, $Dc_{max}$ is larger (e.g. for AAE between 520 nm-880 nm, $Dc_{max} = Dc_1 = 115$ nm, ).

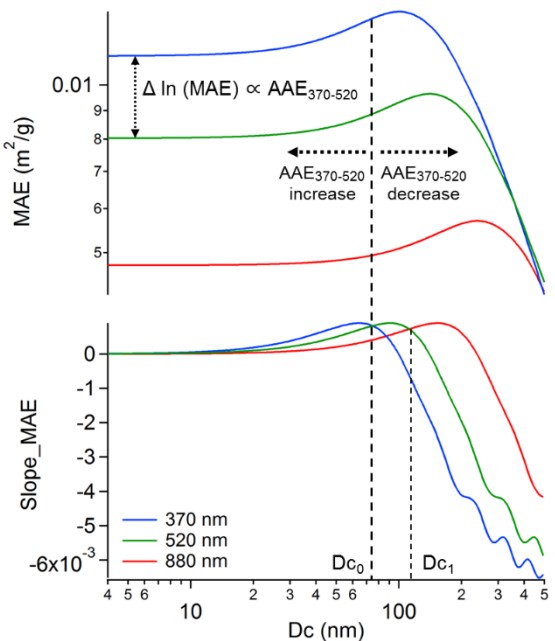

Figure S1. Variation of mass absorption efficiency (MAE) and slope of MAE (slope_MAE) vs. particle diameter (Dc) at 370 nm ($\lambda_1$), 520 nm ($\lambda_2$) and 880 nm for single pure black carbon (BC) at different Dc.

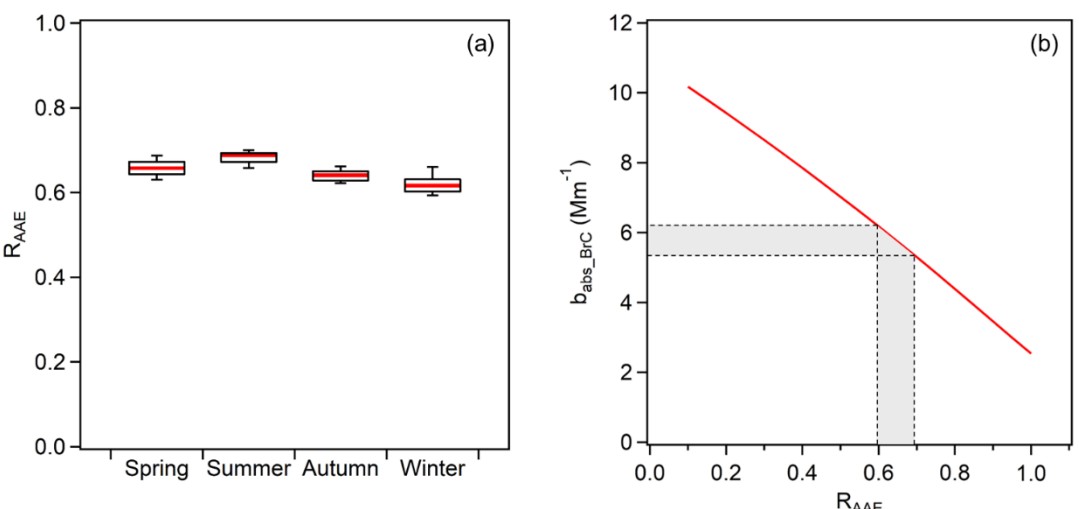

**Figure S2. (a)** Box plot of $R_{AAE}$ in four seasons calculated based on SP2 data; (b) the relationship between different adopted $R_{AAE}$ value and calculated overall mean $b_{abs\_BrC}$. The dash lines of $R_{AAE}$ = 0.60 and 0.69 are 5th and 95th percentile of $R_{AAE}$ data calculated from SP2. The grey area in Y-axis therefore represents the uncertainty range of $b_{abs\_BrC}$

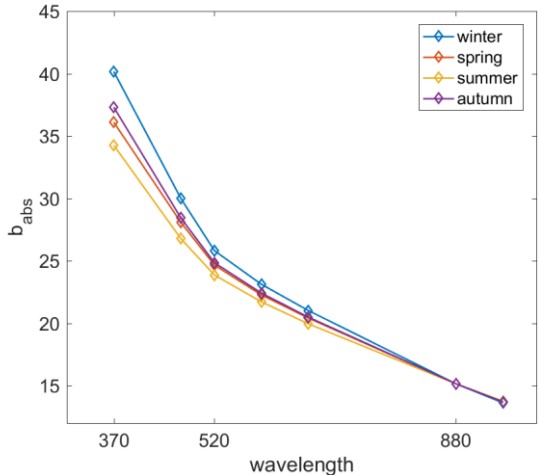

**Figure S3.** Scatter plot of seasonal mean $b_{abs}$ from Aethalometer, data points are normalized using $b_{abs}$ 880 nm

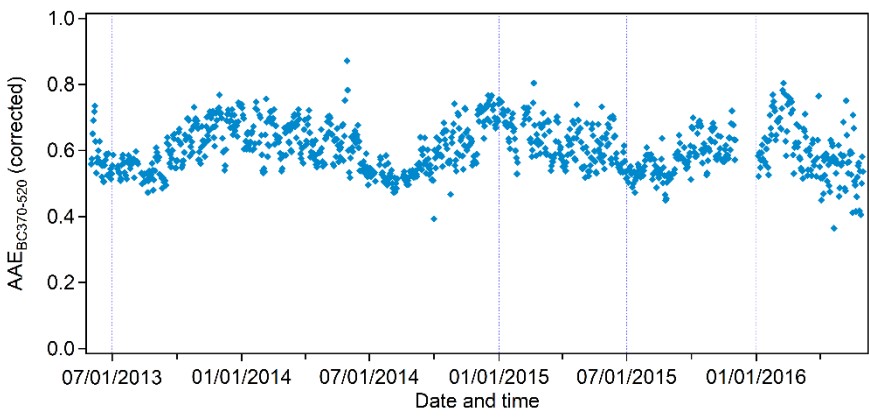

5 **Figure S4.** Time series of corrected $AAE_{BC}$ at 370-520 nm

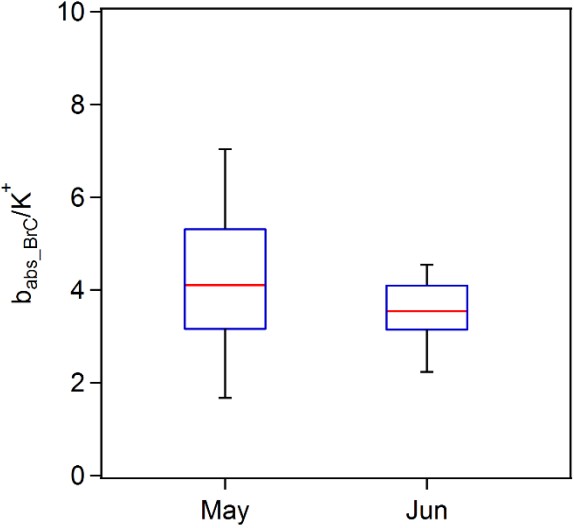

**Figure S5.** Significant difference result of $b_{abs\_BrC}/K^+$ in May and June (data is all from the year 2014)

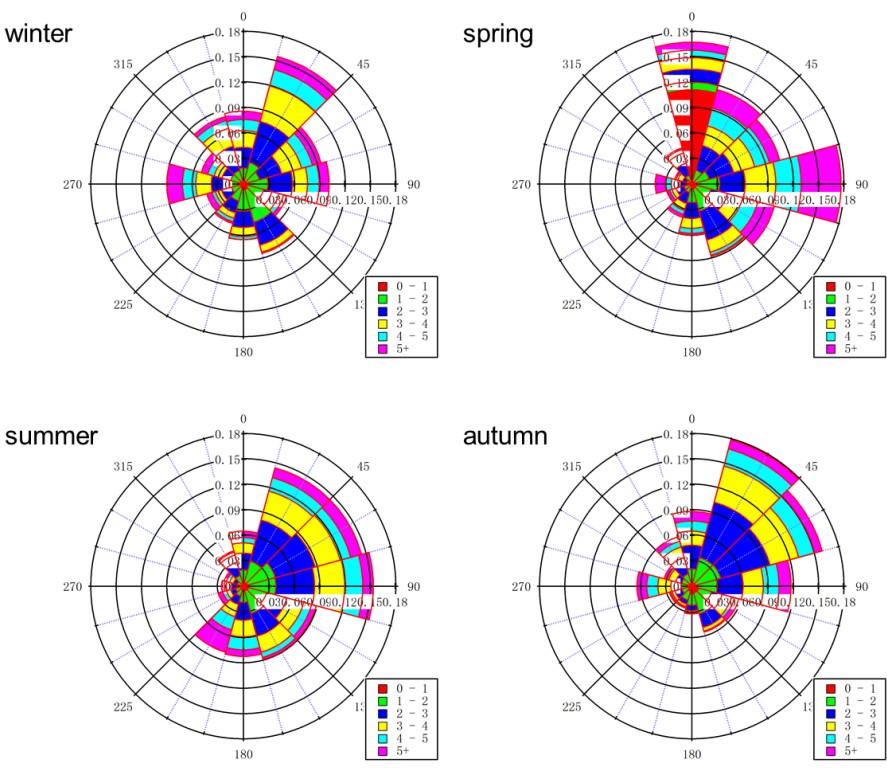

**Figure S6.** Wind roses at the SORPES station in four seasons

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
