# Peer review of "Light absorption of brown carbon in eastern China based on 3year multi-wavelength aerosol optical property observations and an improved Absorption Ångström exponent segregation method"

_Atmospheric Chemistry and Physics, 2018_

## Referee Comment (RC1) · Anonymous Referee #1 · 2 Mar 2018

Wang and coauthors describe the analysis of a three year dataset of multiwavelength aethalometer data in Eastern China using a slightly modified Angstrom exponent approach, coupled with a Mie theory model, to investigate brown carbon optical properties.

Major comments The manuscript is poorly written overall, the original Absorption Angstrom Exponent model development is not discussed or cited, the discussion is

difficult to follow throughout and the findings do not offer much insight into either the sources or physical properties of brown carbon in the region. The major drawback here is the absence of direct measurements of BC particle morphology and coating thickness/composition. Inferring these properties using OCEC data is not appropriate for the Mie theory calculations that follow. Measured particle number-size distribution and coating thickness/composition data would be well suited to investigate the brown carbon properties discussed here. SP2, PAS and SP-AMS instruments, for example would provide more suitable input data. The mixing state of black and brown carbon has been demonstrated to be complex and site-specific, and thus detailed mixing state information is critical as an input for representative Mie theory calculations. If the emphasis is on the aethalometer data processing approach this could indeed be repackaged, however a different journal would likely be a better fit.

---

## Referee Comment (RC2) · Anonymous Referee #2 · 2 Mar 2018

This manuscript improved the previous absorption angstrom exponent segregation method based on Mie-theory simulation and measurement of multi-wavelength aerosol light absorption, by calculating AAEBC at each time step instead of assuming a constant. With the improved method, this study estimated light absorption of BrC and its relative contributions to total light absorption in Yangtze River Delta, China, investigated its seasonal and diurnal variations, and pointed out its dominant sources. However, there are still some major concerns about the technique parts, especially the

uncertainties of the method, which should be considered by the authors before the manuscript could be published in ACP.

General Comments The authors tried to make improvement of aetholometer model method. However, the parameters used here, such as volume, densities and sizes of OC and EC, OM/OC ratios, and morphology and mixing state/compositions of BC particles, which are critical inputs for Mie-theory simulations, were not obtained from direct measurements, but inferred from indirect calculations based on some assumptions. Therefore, my biggest concern is about the uncertainties related to these assumptions and calculations. It would be better to add a part to notify the uncertainties and estimate the uncertainties if possible.

Specific comments 1) Line 28 on page 1: It is better to defined "babs_BrC" as "light absorption coefficient of BrC", with unit as Mm-1. 2) Line 26-27 on page 4 and line 1-6 on Page 5: There are still some brown carbon related studies conducted in China especially in north China such as in Beijing, although these studies were not based on AAE method. 3) Line 10 on page 6: It is still not clear, why 520 nm but not other wavelength was chosen for the following calculations. 4) Line 13 on page 6: As mentioned in the manuscript, some factors might influence the accuracy of the Aethalometer measurement, such as relative humidity especially when the air was not dried prior to sampling. So is RH considered when the authors did the correction for the Aethalometer observation data? 5) Line 24 on page 6: Please add references for the laser absorbance technique for pyrolytically generated carbon correction, or at least give a brief introduction here. 6) Line 26-28 on page 6: Compositions and properties of fine particulate matter might be more complicated in China than other regions. However, the references Aiken et al., 2008 and Pitchford t al., 2007 were both conducted outside China several years ago. So, is it possible to cite some recent literatures especially those conducted in China? And please add a reference for density of EC (1.8 g/cm3). 7) Line 8 on page 7: Please clarify the potential influence of dust, another important light absorber in the atmosphere, and explain how to exclude it. 8) Eq 2: How about the

influence of other light-absorbing substances, and the coating thickness/composition of black carbon? 9) Line 4-5 on page 8: It would be better to point out major factors which might influence the AAE values or light absorption properties of BC, before the sentence "Therefore, it is essential to firstly evaluate . . .." 10) Line 8 on page 10: The current display format of Figure 3 is not quite powerful for the statement of "especially in less polluted periods", as many negative values also occurred in higher polluted days, with even lower negative values. 11) The findings of this study were consistent with previous studies. It is better to refer to or compared with more previous studies. For example, some studies have also pointed out that coal combustion was one of the major sources of brown carbon in north China in winter.

---

## Referee Comment (RC3) · D. Liu (Referee) · 5 Mar 2018

This study provides an improved approach on deriving the brown carbon absorption from AE31 measurement, and highlights the importance in using the proper AAE to extrapolate the BC absorption from longer to shorter wavelength. If the technical part of this study could be more convincible, then it could be considered for publication. I would suggest to improve the technical part by considering the following points.

[Figure]

1.The issue of deriving the coating content from OC/EC measurement is not just from the uncertain OM/OC ratio, but also that many of the OM may not contain BC, i.e. externally mixed, the OC/EC method would tend to largely overestimate the coating associated with BC. You could use some external results to estimate this.

2. The crucial results here are from the AE31. How it has been corrected is important. It is a challenge to get the proper result from this instrument (especially at shorter wavelength). Though this may have been done in your previous publications, but it's worthy to mention here, e.g. how you get the multiple scattering and scattering correction, and how you have corrected these at different wavelength, and these may substantially affect the derived AAE because these corrections rely on the AAE as well.

3. The BC core size is not constant, the BC from open biomass burning or domestic solid fuel burning has a larger core than from traffic (Schwarz et al., 2008) (Liu et al., 2014). As you have already raised, the larger core will have a lower AAE. Some external data could be used to make more constrains for biomass burning BC.

4. The ideal approach would be combing with the SP2 measurement, such as a similar study (Liu et al., 2015), however the advantage of this study is the long-term measurement using the less-cost instrumentation, and different contributions such as open biomass burning or residential burning occurred in different months, which is interesting. If by somehow, this study could benefit by constraining some of the inputs from the existing information and doing sensitivity test.

Other comments: -It is better to show the location of the experimental site and the surrounding major emissions.

-Equation 4 and 5 could be merged into one, it would be useful to show the Rs-1 in time series or monthly variation.

-Please give the refractive index you used for BC, clear coatings and brown coatings in the main plot legends.

-the Cl-/EC, as an indicator of coal combustion, needs more reference

-would you be able to derive the MAE of brown carbon at different months, which will be very interesting. It looks the higher Babs/K+ ratio possibly means the Dec. BrC had a higher absorption efficiency maybe?

Schwarz, J., Gao, R., Spackman, J., Watts, L., Thomson, D., Fahey, D., Ryerson, T., Peischl, J., Holloway, J., and Trainer, M.: Measurement of the mixing state, mass, and optical size of individual black carbon particles in urban and biomass burning emissions, Geophys. Res. Lett., 35, 2008.

Liu, D., Allan, J., Young, D., Coe, H., Beddows, D., Fleming, Z., Flynn, M., Gallagher, M., Harrison, R., and Lee, J.: Size distribution, mixing state and source apportionment of black carbon aerosol in London during wintertime, Atmos. Chem. Phys., 14, 10061-10084, 2014.

Liu, D., Taylor, J. W., Young, D. E., Flynn, M. J., Coe, H., and Allan, J. D.: The effect of complex black carbon microphysics on the determination of the optical properties of brown carbon, Geophys. Res. Lett., 42, 613-619, 2015.
* * *

---

## Author Comment (AC2) · 6 May 2018

**Wang, J., Nie, W., Cheng, Y., Shen, Y., Chi, X., Wang, J., Huang, X., Xie, Y., Sun, P., Xu, Z., Qi, X., Su, H., and Ding, A.: Light absorption of brown carbon in eastern China based on 3-year multi-wavelength aerosol optical property observations at the SORPES station and an improved Absorption Ångstrom exponent segregation method, Atmos. Chem. Phys. Discuss., https://doi.org/10.5194/acp-2018-49, in review, 2018.**

**Replies to reviewers' comments**

**Overview**

The authors thank the reviewers for constructive comments, they helped improving the paper. We have replied to all questions raised by the reviewer. The major changes to the paper are that the we have

- reorganized the AAE method part and revised corresponding figures

- added measurement data from single particle soot photometer (SP2) at same site to support calculation

- conducting Mie-simulation using BC size distribution and mixing state measured by SP2 to derive $AAE_{BC}$ instead of taking a fixed typical BC core size distribution and analyzed the uncertainties of the method

- added more review on AAE segregation and BrC definition

**Detailed replies to Anonymous Referee #2**

General Comments The authors tried to make improvement of Aethalometer model method. However, the parameters used here, such as volume, densities and sizes of OC and EC, OM/OC ratios, and morphology and mixing state/compositions of BC particles, which are critical inputs for Mie-theory simulations, were not obtained from direct measurements, but inferred from indirect calculations based on some assumptions. Therefore, my biggest concern is about the uncertainties related to these assumptions and calculations. It would be better to add a part to notify the uncertainties and estimate the uncertainties if possible.

*Response: Thanks for the valuable comment. To avoid the uncertainties caused by assumed BC size distribution, we reorganized our manuscript by using available BC size distribution and mixing state measured by single particle soot photometer (SP2) at the SORPES station during the study period instead of using empirical BC size distribution from previous studies. We found that our previous method is still available but we modified the method by combining SP2 data in the revision. The contribution of BrC gets larger with new parameters. Also, in the revised version we estimate the uncertainties of the calculation and update the method and result parts.*

**Specific comments**

1) Line 28 on page 1: It is better to defined "babs_BrC" as "light absorption coefficient of BrC", with unit as Mm-1.

*Response: Thanks for the suggestion. We will revise it accordingly.*

2) Line 26-27 on page 4 and line 1-6 on Page 5: There are still some brown carbon related studies conducted in China especially in north China such as in Beijing, although these studies were not based on AAE method.

*Response: Thanks for the suggestion. We will add some relevant brown carbon studies in North China in the discussion and include these references in the revised version.*

3) Line 10 on page 6: It is still not clear, why 520 nm but not other wavelength was chosen for the following calculations.

*Response: Thanks for the comment. As mentioned in the manuscript, the measurement wavelengths of aethalometer are 370 nm, 470 nm, 520 nm, 590 nm, 660 nm, 880 nm, and 950 nm. Measurement data at the wavelength of 880 nm is chosen because light absorption at this wavelength normally represents the BC absorption (Virkkula et al., 2015). To calculate $AAE_{BC}$ at the wavelength range where BC is the dominant absorption component, the wavelength near 880 nm is better to be used. However, the response of 590 nm and 660 nm data may be affected by the presence of interfering materials such as hematite mineral dust and tobacco smoke (user manual of aethalometer AE-31, Hansen and Schnell, 2005). To avoid the potential impact, we use 520 nm data for the calculations. We will revise the text to further clarify this point in the revision.*

4) Line 13 on page 6: As mentioned in the manuscript, some factors might influence the accuracy of the Aethalometer measurement, such as relative humidity especially when the air was not dried prior to sampling. So is RH considered when the authors did the correction for the Aethalometer observation data?

*Response: Thanks for the comment. We agree that relative humidity would influence the accuracy of Aethalometer measurement. In our study, RH is considered in both measurement and Aethalometer data correction processes. The impacts of RH on Aethalometer data are mainly from hygroscopic growth of scattering particles and the hydrophilic membrane under the filter. Firstly, we have added the detailed description of aerosol scattering measurement and the correction of Aethalometer data. Since the relative humidity can make influence on aerosol extinction, especially scattering, we installed an external heater prior to the measurement to prevent condensation and maintain the sample air humidity lower than 50% for the most*

*of time. For data with RH higher than 50%, we did the correction for it. Detailed description is shown in the revised manuscript. Secondly, the bias of Aethalometer data caused by the hydrophilic membrane under the filter only occurs during rapid RH changes (Arnott et al., 2003). For slow RH changes, the humidification factor is compensated when calculating the attenuation change. Compared to these two factors, the direct impact of RH on light absorption is negligible (Redemann et al., 2001).*

5) Line 24 on page 6: Please add references for the laser absorbance technique for pyrolytically generated carbon correction, or at least give a brief introduction here.

*Response: Thanks for the comment. After analyzing SP2 data, we have reorganized the method and result part as well as the corresponding figures, and OCEC data is no longer used in the Figures. Thus, the description of OCEC measurement part will be removed in the revised manuscript.*

6) Line 26-28 on page 6: Compositions and properties of fine particulate matter might be more complicated in China than other regions. However, the references Aiken et al., 2008 and Pitchford t al., 2007 were both conducted outside China several years ago. So, is it possible to cite some recent literatures especially those conducted in China? And please add a reference for density of EC (1.8 g/cm3).

*Response: Thanks for the comment. Indeed, observation of OC/EC (Cheng et al., 2010;Cheng et al., 2011) are widely used in China. As mentioned in last response, we no longer use OC/EC data in the revised manuscript. The comparison of characteristics of BrC in Nanjing and Beijing (Yan et al., 2015;Cheng et al., 2016;Yan et al., 2017) are supplied in this study, as mentioned in the response to Comment 2).*

7) Line 8 on page 7: Please clarify the potential influence of dust, another important light absorber in the atmosphere, and explain how to exclude it.

*Response: Thanks for the comment. Firstly, dust is abundant in coarse mode (Seinfeld and Pandis, 2006;Hu et al., 2013). The mass fraction of dust is low in $PM_{2.5}$ (Wang et al., 2017) except during dust storm episode. We used a $PM_{2.5}$ cutter on the inlet of Aethalometer to avoid the impact of coarse mode particles. Moreover, the mass absorption efficiency (MAE) of dust is around 1% of MAE of BC (Clark et al., 2016). Thus, the contribution of dust on $PM_{2.5}$ light absorption should be small. We will mentioned this point in the revised manuscript.*

[Figure]

***Figure 1.*** *Time series of the concentration of $Ca^{2+}$, a tracer to dust, OC and $b_{abs\_BrC}$ measured at the SORPES station. The figure shows that the calculated light absorption of BrC was less influenced by dust.*

8) Eq 2: How about the influence of other light-absorbing substances, and the coating thickness/composition of black carbon?

***Response****: Thanks for the comment. As mentioned in the introduction, BC, dust, and BrC have been considered as dominant contributors to aerosol absorption. In our revised manuscript, the impact of coating thickness will be considered in both SP2 data analysis and $AAE_{520-880}$. We calculate $AAE_{BC520-880}$ and $AAE_{BC370-520}$ from SP2 data using core-shell Mie model. Coating thickness derived from SP2 is included in the calculation. In long-term observation, $AAE_{520-880}$ is calculated based on data from the Aethalometer AE31.*

9) Line 4-5 on page 8: It would be better to point out major factors which might influence the AAE values or light absorption properties of BC, before the sentence "Therefore, it is essential to firstly evaluate . . .."

***Response****: Thanks for the comment. We will modify this description in the revision.*

10) Line 8 on page 10: The current display format of Figure 3 is not quite powerful for the statement of "especially in less polluted periods", as many negative values also occurred in higher polluted days, with even lower negative values.

***Response****: Thanks for the comment. Yes, this expression is not very clear here. More precise expression should be 'when light absorption of BrC is low, more negative values occur by using the assumption of $AAE_{BC} = 1$' (shown in Figure 4 in revised manusrcipt). We will change this sentence in the revised manuscript.*

11) The findings of this study were consistent with previous studies. It is better to refer to or compared with more previous studies. For example, some studies have also pointed out that coal combustion was one of the major sources of brown carbon in north China in winter.

[Figure]

*Figure 1. Averaged retroplume for air masses with highest 25% and lowest 25% of $b_{abs\_BrC}$ at the SORPES station in December. The results show that long-range transport from the North China Plain had limited influence on BrC at the station.*

***Response****: Thanks for the suggestion. We compared characteristics of BrC in Nanjing and Beijing (Yan et al., 2015;Cheng et al., 2016;Yan et al., 2017). Studies conducted in Beijing have also suggested the important contribution of residential biofuel and open biomass burning on BrC in winter and summer, respectively (Yan et al., 2015;Cheng et al., 2016;Fu et al., 2013), which are consistent with our conclusions. A recent study reports that coal combustion is an important source of BrC in northern China, especially in winter (Yan et al., 2017).We will discuss these factors in the revision.*

**References:**

Arnott, W. P., Moosmüller, H., Sheridan, P. J., Ogren, J. A., Raspet, R., Slaton, W. V., Hand, J. L., Kreidenweis, S. M., and Collett, J. L.: Photoacoustic and filter-based ambient aerosol light absorption measurements: Instrument comparisons and the role of relative humidity, Journal of Geophysical Research: Atmospheres (1984–2012), 108, AAC 15-11, 10.1029/2002JD002165, 2003.

Cheng, Y., He, K. B., Duan, F. K., Zheng, M., Ma, Y. L., Tan, J. H., and Du, Z. Y.: Improved measurement of carbonaceous aerosol: evaluation of the sampling artifacts and inter-comparison of the thermal-optical analysis methods, Atmos. Chem. Phys. , 10, 8533-8548, 10.5194/acp-10-8533-2010, 2010.

Cheng, Y., He, K. B., Zheng, M., Duan, F. K., Du, Z. Y., Ma, Y. L., Tan, J. H., Yang, F. M., Liu, J. M., Zhang, X. L., Weber, R. J., Bergin, M. H., and Russell, A. G.: Mass absorption efficiency of elemental carbon and water-soluble organic carbon in Beijing, China, Atmos. Chem. Phys., 11, 11497-11510, 10.5194/acp-11-11497-2011, 2011.

Cheng, Y., He, K.-b., Du, Z.-y., Engling, G., Liu, J.-m., Ma, Y.-l., Zheng, M., and Weber, R. J.: The characteristics of brown carbon aerosol during winter in Beijing, Atmos. Environ. , 127, 355-364, 10.1016/j.atmosenv.2015.12.035, 2016.

Clark, C. J., Schofield, S. P., Gomez, H. L., and Davies, J. I.: An empirical determination of the dust mass absorption coefficient, κ d, using the Herschel Reference Survey, Monthly Notices of the Royal Astronomical Society, 459, 1646-1658, 2016.

Fu, X., Wang, S., Zhao, B., Xing, J., Cheng, Z., Liu, H., and Hao, J.: Emission inventory of primary pollutants and chemical speciation in 2010 for the Yangtze River Delta region, China, Atmos. Environ. , 70, 39-50, https://doi.org/10.1016/j.atmosenv.2012.12.034, 2013.

Hu, X., Ding, Z., Zhang, Y., Sun, Y., Wu, J., Chen, Y., and Lian, H.: Size distribution and source apportionment of airborne metallic elements in Nanjing, China, Aerosol Air Qual. Res, 13, 1796-1806, 2013.

Redemann, J., Russell, P. B., and Hamill, P.: Dependence of aerosol light absorption and single-scattering albedo on ambient relative humidity for sulfate aerosols with black carbon cores, Journal of Geophysical Research: Atmospheres, 106, 27485-27495, doi:10.1029/2001JD900231, 2001.

Seinfeld, J., and Pandis, S.: Atmospheric chemistry and physics. Hoboken, in, NJ: Wiley, 2006.

Virkkula, A., Chi, X., Ding, A., Shen, Y., Nie, W., Qi, X., Zheng, L., Huang, X., Xie, Y., Wang, J., Petäjä, T., and Kulmala, M.: On the interpretation of the loading correction of the aethalometer, Atmos. Meas. Tech., 8, 4415-4427, 10.5194/amt-8-4415-2015, 2015.

Wang, J., Zhao, B., Wang, S., Yang, F., Xing, J., Morawska, L., Ding, A., Kulmala, M., Kerminen, V. M., Kujansuu, J., Wang, Z., Ding, D., Zhang, X., Wang, H., Tian, M., Petaja, T., Jiang, J., and Hao, J.: Particulate matter pollution over China and the effects of control policies, Sci Total Environ, 584-585, 426-447, 10.1016/j.scitotenv.2017.01.027, 2017.

Yan, C., Zheng, M., Sullivan, A. P., Bosch, C., Desyaterik, Y., Andersson, A., Li, X., Guo, X., Zhou, T., Gustafsson, Ö., and Collett, J. L.: Chemical characteristics and light-absorbing property of water-soluble organic carbon in Beijing: Biomass burning contributions, Atmospheric Environment, 121, 4-12, https://doi.org/10.1016/j.atmosenv.2015.05.005, 2015.

Yan, C., Zheng, M., Bosch, C., Andersson, A., Desyaterik, Y., Sullivan, A. P., Collett, J. L., Zhao, B., Wang, S., He, K., and Gustafsson, O.: Important fossil source contribution to brown carbon in Beijing during winter, Sci Rep, 7, 43182, 10.1038/srep43182, 2017.

---

## Author Comment (AC3) · 6 May 2018

**Wang, J., Nie, W., Cheng, Y., Shen, Y., Chi, X., Wang, J., Huang, X., Xie, Y., Sun, P., Xu, Z., Qi, X., Su, H., and Ding, A.: Light absorption of brown carbon in eastern China based on 3-year multi-wavelength aerosol optical property observations at the SORPES station and an improved Absorption Ångstrom exponent segregation method, Atmos. Chem. Phys. Discuss., https://doi.org/10.5194/acp-2018-49, in review, 2018.**

**Replies to reviewers' comments**

**Overview**

The authors thank the reviewers for constructive comments, they helped improving the paper. We have replied to all questions raised by the reviewer. The major changes to the paper are that the we have

- reorganized the AAE method part and revised corresponding figures

- added measurement data from single particle soot photometer (SP2) at same site to support calculation

- conducting Mie-simulation using BC size distribution and mixing state measured by SP2 to derive $AAE_{BC}$ instead of taking a fixed typical BC core size distribution and analyzed the uncertainties of the method

- added more review on AAE segregation and BrC definition

**Detailed replies to Anonymous Referee #3**

General Comments This study provides an improved approach on deriving the brown carbon absorption from AE31 measurement, and highlights the importance in using the proper AAE to extrapolate the BC absorption from longer to shorter wavelength. If the technical part of this study could be more convincible, then it could be considered for publication. I would suggest to improve the technical part by considering the following points.

*Response: Thanks for the valuable comment. To avoid the uncertainties caused by assumed BC size distribution, we will reorganized our manuscript by using BC size distribution and mixing state measured by single particle soot photometer (SP2) at the SORPES station instead of using empirical BC size distribution from previous studies. We found that our previous method is still available but we modified the method by combining SP2 data in the revision. The contribution of BrC gets larger with new parameters. Also, in the revised version we estimate the uncertainties of the calculation and update the method and result parts.*

**Specific comments**

1) The issue of deriving the coating content from OC/EC measurement is not just from the uncertain OM/OC ratio, but also that many of the OM may not contain BC, i.e. externally mixed, the OC/EC method would tend to largely overestimate the coating associated with BC. You could use some external results to estimate this.

*Response: Thanks for the comment. We agree that the mixing state of BC has impact on BC optical properties. We will use available SP2 data during the study to get mixing state and size of BC and revised the calculation supported by SP2 results. OCEC data are no longer used in this study. Detailed description will be presented in the revision.*

2) The crucial results here are from the AE31. How it has been corrected is important. It is a challenge to get the proper result from this instrument (especially at shorter wavelength). Though this may have been done in your previous publications, but it's worthy to mention here, e.g. how you get the multiple scattering and scattering correction, and how you have corrected these at different wavelength, and these may substantially affect the derived AAE because these corrections rely on the AAE as well.

*Response:*

*Thanks for the comment. For Aethalometer data correction, we implemented the correction algorithm presented by Collaud Coen et al. (2010). $b_{abs}$ at wavelength λ is corrected for filter-loading effect, scattering effect and multiple scattering effect, with equation shown as*

$$b_{abs}(\lambda) = \frac{b_{ATN} - s' \cdot b_{scat}}{C_{ref} \cdot R}$$

*where R is the function for filter-loading correction and s' represents the fraction of scattering coefficient resulting in ATN change (Shen et al., 2018). $C_{ref}$ is the multiple scattering correction factor, which is set to be 4.26 according to Collaud Coen et al. (2010). Detailed calculations of R and s' can be found in Collaud Coen et al. (2010). The comparison of different Aethalometer corrections (Saturno et al., 2017) shows that AAE derived by Collaud Coen correction algorithm agrees well with that from multi-wavelength reference measurement, proving the reliable AAE values calculated from this correction. Saturno et al. (2017) also proves that Collaud Coen correction shows a good performance in obtaining absorption coefficients at 370 nm, which is the critical wavelength in BrC segregation. We will add more description of Aethalometer data correction in the revised manuscript.*

3) The BC core size is not constant, the BC from open biomass burning or domestic solid fuel burning has a larger core than from traffic (Schwarz et al., 2008) (Liu et al., 2014). As you have already raised, the larger core will have a lower AAE. Some external data could be used to make more constrains for biomass burning BC.

*Response: Thanks for the comment. We agree that BC core size can vary under influence of different dominant sources. As suggested, to derive AAE$_{BC}$, we will use real-time core size and coating thickness of BC measured by SP2 instead of taking a fixed typical BC core size distribution.*

4) The ideal approach would be combing with the SP2 measurement, such as a similar study (Liu et al., 2015), however the advantage of this study is the long-term measurement using the less-cost instrumentation, and different contributions such as open biomass burning or residential burning occurred in different months, which is interesting. If by somehow, this study could benefit by constraining some of the inputs from the existing information and doing sensitivity test.

*Response: Thanks for the comment. We will combine SP2 measurement to modify the method. Since there are lots of historical observation data conducted without SP2, it is worth to try and find a way of tracing back light absorption of BrC with satisfactory uncertainty range. At this site, firstly, we will calculate AAE$_{BC}$ at short and long wavelength range based on available SP2 data during the study period, and analyzed their variation ranges. Uncertainty and sensitivity of this method will also be added in the revised manuscript.*

5) It is better to show the location of the experimental site and the surrounding major emissions.

*Response: Thanks for the comment. The map and emission character surrounding the site have been given in many of our previous works, e.g. (Ding et al., 2013;Ding et al., 2016). Here we would like to cite these reference and describe the character in the text. The location of the SORPES site will also be given in several figures in the revision.*

6) Equation 4 and 5 could be merged into one, it would be useful to show the Rs-1 in time series or monthly variation.

*Response: Thanks for the suggestion. R$_{AAE}$ will be used in revised manuscript instead of R$_{s-l}$ for better*

*understanding. We will add the definition of $R_{AAE}$ in the text instead of showing as Eq. 4. The Eq. 5 in the original manuscript will be numbered as Eq. 6 in the revised manuscript. We will also show the seasonal variation of $R_{AAE}$ in the revision.*

7) Please give the refractive index you used for BC, clear coatings and brown coatings in the main plot legends.

**Response**: *Thanks for the suggestion. The refractive index (RI) of BC core was set to be 1.56+0.47i according to Dalzell and Sarofim (1969) and RI was 1.52+0i for clear shell (Pitchford et al., 2007). These values will also be listed in the revised manuscript and Figure 1. The brown coating case will not be discussed in our revised manuscript, due to large uncertainties on brown coating's refractive index.*

8) Cl-/EC, as an indicator of coal combustion, needs more reference, would you be able to derive the MAE of brown carbon at different months, which will be very interesting. It looks the higher Babs/K+ ratio possibly means the Dec. BrC had a higher absorption efficiency maybe?

**Response**: *Thanks for the comment. After revision of the method and related results, we will also modify the source analysis part in the revised manuscript. Detailed description can be added. About $MAE_{BrC}$, thanks for this interesting suggestion. However, mass of BrC was not measured directly at this site and we can only use OC data to convert the mass of OC to mass of BrC. Since BrC mass fraction in OM may change over time and case due to different dominant sources, especially in China where the emission profile is complicated, it is hard to differentiate the change of $MAE_{BrC}$ and BrC mass fraction. In the future, we will try to calculate $MAE_{BrC}$ by using water soluble OC (WSOC) measurement and have a detail study focusing on this topic. Thanks again for your valuable suggestion.*

**References:**

Collaud Coen, M., Weingartner, E., Apituley, A., Ceburnis, D., Fierz-Schmidhauser, R., Flentje, H., Henzing, J., Jennings, S. G., Moerman, M., and Petzold, A.: Minimizing light absorption measurement artifacts of the Aethalometer: evaluation of five correction algorithms, Atmos. Meas. Tech., 3, 457-474, 2010.

Dalzell, W. H., and Sarofim, A. F.: Optical Constants of Soot and Their Application to Heat-Flux Calculations, J. Heat Transfer 91, 100-104, 10.1115/1.3580063, 1969.

Ding, A., Fu, C., Yang, X., Sun, J., Petäjä, T., Kerminen, V.-M., Wang, T., Xie, Y., Herrmann, E., and Zheng, L.: Intense atmospheric pollution modifies weather: a case of mixed biomass burning with fossil fuel combustion pollution in eastern China, Atmos. Chem. Phys. , 13, 10545-10554, 2013.

Ding, A., Nie, W., Huang, X., Chi, X., Sun, J., Kerminen, V.-M., Xu, Z., Guo, W., Petäjä, T., Yang, X., Kulmala, M., and Fu, C.: Long-term observation of air pollution-weather/climate interactions at the SORPES station: a review and

outlook, Frontiers of Environmental Science & Engineering, 10, 15-, 10.1007/s11783-016-0877-3, 2016.

Pitchford, M., Malm, W., Schichtel, B., Kumar, N., Lowenthal, D., and Hand, J.: Revised Algorithm for Estimating Light Extinction from IMPROVE Particle Speciation Data, J. Air Waste manage., 57, 1326-1336, 10.3155/1047-3289.57.11.1326, 2007.

Saturno, J., Pöhlker, C., Massabò, D., Brito, J., Carbone, S., Cheng, Y., Chi, X., Ditas, F., Hraběde Angelis, I., Morán-Zuloaga, D., Pöhlker, M. L., Rizzo, L. V., Walter, D., Wang, Q., Artaxo, P., Prati, P., and Andreae, M. O.: Comparison of different Aethalometer correction schemes and a reference multi-wavelength absorption technique for ambient aerosol data, Atmos. Meas. Tech., 10, 2837-2850, 10.5194/amt-10-2837-2017, 2017.

Shen, Y., Virkkula, A., Ding, A., Wang, J., Chi, X., Nie, W., Qi, X., Huang, X., Liu, Q., Zheng, L., Xu, Z., Petäjä, T., Aalto, P. P., Fu, C., and Kulmala, M.: Aerosol optical properties at SORPES in Nanjing, east China, Atmos. Chem. Phys. , 18, 5265-5292, 10.5194/acp-18-5265-2018, 2018.

---

## Author Response (AR1)

**Wang, J., Nie, W., Cheng, Y., Shen, Y., Chi, X., Wang, J., Huang, X., Xie, Y., Sun, P., Xu, Z., Qi, X., Su, H., and Ding, A.: Light absorption of brown carbon in eastern China based on 3-year multi-wavelength aerosol optical property observations at the SORPES station and an improved Absorption Ångstrom exponent segregation method, Atmos. Chem. Phys. Discuss., https://doi.org/10.5194/acp-2018-49, in review, 2018.**

**Replies to reviewers' comments**

**Overview**

The authors thank the reviewers for constructive comments, they helped improving the paper. We have replied to all questions raised by the reviewer. The major changes to the paper are:

- added measurement data from single particle soot photometer (SP2) at same site to support calculation

- conducting Mie-simulation using BC size distribution and mixing state measured by SP2 to derive $AAE_{BC}$ instead of taking a fixed typical BC core size distribution and analyzed the uncertainties of the method

- reorganized the method part and revised corresponding figures

- added more reviews on AAE segregation and BrC related discussions

Below the responses are written in cursive letters and the changes to the manuscript are in blue color or shown the number of Page and Line and in red color in revised manuscript.

**Detailed replies to reviewers' comments**

**Detailed replies to Anonymous Referee #1**

Wang and coauthors describe the analysis of a three year dataset of multi-wavelength aethalometer data in Eastern China using a slightly modified Angstrom exponent approach, coupled with a Mie theory model, to investigate brown carbon optical properties.

Major comments: The manuscript is poorly written overall, the original Absorption Angstrom Exponent model development is not discussed or cited.

The discussion is difficult to follow throughout and the findings do not offer much insight into either the sources or physical properties of brown carbon in the region.

The major drawback here is the absence of direct measurements of BC particle morphology and coating

thickness/composition. Inferring these properties using OCEC data is not appropriate for the Mie theory calculations that follow. Measured particle number-size distribution and coating thickness/composition data would be well suited to investigate the brown carbon properties discussed here. SP2, PAS and SP-AMS instruments, for example would provide more suitable input data. The mixing state of black and brown carbon has been demonstrated to be complex and site-specific, and thus detailed mixing state information is critical as an input for representative Mie theory calculations. If the emphasis is on the aethalometer data processing approach this could indeed be repackaged, however a different journal would likely be a better fit.

**Response:** *We thank the Referee for these valuable comments, which did help us improve the previous version of this manuscript. Following the referee's suggestions, we have removed the OC/EC data in the revised manuscript and added available SP2 measurement to improve the calculation method. We have reorganized the manuscript with updated results and cited relevant literatures as possible as we can. We think that the updated data and results will offer insights into the characteristics and sources of brown carbon in eastern China, a region with heavily polluted air quality but less investigations on brown carbon.*

**Detailed replies to Anonymous Referee #2**

General Comments The authors tried to make improvement of Aethalometer model method. However, the parameters used here, such as volume, densities and sizes of OC and EC, OM/OC ratios, and morphology and mixing state/compositions of BC particles, which are critical inputs for Mie-theory simulations, were not obtained from direct measurements, but inferred from indirect calculations based on some assumptions. Therefore, my biggest concern is about the uncertainties related to these assumptions and calculations. It would be better to add a part to notify the uncertainties and estimate the uncertainties if possible.

*Response: Thanks for the valuable comment. To avoid the uncertainties caused by assumed BC size distribution, we reorganized our manuscript by using BC size distribution and mixing state measured by single particle soot photometer (SP2) at the SORPES station instead of using empirical BC size distribution from previous studies. We found that our previous method is still available but we modified the method by combining SP2 data in the revision. The contribution of BrC gets larger with new parameters. Also, in the revised version we estimated the uncertainties of the calculation and updated the method and result parts. Detailed revision can be found in marked-up revised manuscript.*

**Specific comments**

1) Line 28 on page 1: It is better to defined "babs_BrC" as "light absorption coefficient of BrC", with unit as Mm-1.

*Response: Thanks for the suggestion. We have revised it accordingly shown in Page 1, Line 27 in revised manuscript.*

> *'...The anuual average light absorption coefficient of BrC ($b_{abs\_BrC}$) at 370 nm was 6.3 Mm$^{-1}$ at the SORPES station....'*

2) Line 26-27 on page 4 and line 1-6 on Page 5: There are still some brown carbon related studies conducted in China especially in north China such as in Beijing, although these studies were not based on AAE method.

*Response: Thanks for the suggestion. We added some relevant brown carbon studies in North China in the discussion and included these references in the revised version. We compared characteristics of BrC in Nanjing and Beijing (Yan et al., 2015;Cheng et al., 2016;Yan et al., 2017). Studies conducted in Beijing have also*

*suggested the important contribution of residential biofuel and open biomass burning on BrC in winter and summer, respectively (Yan et al., 2015;Cheng et al., 2016;Fu et al., 2013), which are consistent with our conclusions. A recent study reports that coal combustion is an important source of BrC in northern China, especially in winter (Yan et al., 2017). The discussions are added in Page 15, Line 13-15 in revised manuscript.*

3) Line 10 on page 6: It is still not clear, why 520 nm but not other wavelength was chosen for the following calculations.

***Response***: *Thanks for the comment. As mentioned in the manuscript, the measurement wavelengths of aethalometer are 370 nm, 470 nm, 520 nm, 590 nm, 660 nm, 880 nm, and 950 nm. Measurement data at the wavelength of 880 nm is chosen because light absorption at this wavelength normally represents the BC absorption (Virkkula et al., 2015). To calculate AAE$_{BC}$ at the wavelength range where BC is the dominant absorption component, the wavelength near 880 nm is better to be used. However, the response of 590 nm and 660 nm data may be affected by the presence of interfering materials such as hematite mineral dust and tobacco smoke (user manual of aethalometer AE-31, Hansen and Schnell, 2005). To avoid the potential impact, we use 520 nm data for the calculations. We have added the description in Page 6, Line 15-20 in revised manuscript.*

> *'Regarding to the wavelength dependence analysis of BC and BrC, the wavelength near 880 nm is better for the calculation of AAE of BC because BC is the dominant absorption components at that wavelength range. However, the response of the 590 nm and 660 nm data may be affected by the presence of interfering materials such as hematite mineral dust and tobacco smoke (user manual of aethalometer AE-31, Hansen and Schnell, 2005), hence 520 nm data was used for the following calculations.'*

4) Line 13 on page 6: As mentioned in the manuscript, some factors might influence the accuracy of the Aethalometer measurement, such as relative humidity especially when the air was not dried prior to sampling. So is RH considered when the authors did the correction for the Aethalometer observation data?

***Response***: *Thanks for the comment. We agree that relative humidity would influence the accuracy of Aethalometer measurement. In our study, RH is considered in both measurement and Aethalometer data correction processes. The impacts of RH on Aethalometer data are mainly from hygroscopic growth of scattering particles and the hydrophilic membrane under the filter. Firstly, we have added the detailed*

*description of aerosol scattering measurement and the correction of Aethalometer data. Since the relative humidity can make influence on aerosol extinction, especially scattering, we installed an external heater prior to the measurement to prevent condensation and maintain the sample air humidity lower than 50% for the most of time. For data with RH higher than 50%, we did the correction for it. Detailed description is shown in the revised manuscript. Secondly, the bias of Aethalometer data caused by the hydrophilic membrane under the filter only occurs during rapid RH changes (Arnott et al., 2003). For slow RH changes, the humidification factor is compensated when calculating the attenuation change. Compared to these two factors, the direct impact of RH on light absorption is negligible (Redemann et al., 2001). We have revised this part in Page 6, Line 20-Page 7, Line 19 in revised manuscript.*

5) Line 24 on page 6: Please add references for the laser absorbance technique for pyrolytically generated carbon correction, or at least give a brief introduction here.

*Response: Thanks for the comment. After analyzing SP2 data, we have reorganized the method and result part as well as the corresponding figures, and OCEC data is no longer used in the Figures. Thus, the description of OCEC measurement part has been removed from the revised manuscript, but it can still be mentioned here that Thermo-Optical Transmittance (TOT) protocol is adopted for pyrolysis correction.*

6) Line 26-28 on page 6: Compositions and properties of fine particulate matter might be more complicated in China than other regions. However, the references Aiken et al., 2008 and Pitchford et al., 2007 were both conducted outside China several years ago. So, is it possible to cite some recent literatures especially those conducted in China? And please add a reference for density of EC (1.8 g/cm3).

*Response: Thanks for the comment. Indeed, observation of OC/EC (Cheng et al., 2010;Cheng et al., 2011) are widely used in China. As mentioned in last response, we no longer use OC/EC data. SP2 data is used in the revision and the density of BC is assumed to be 1.8 g cm$^{-3}$ (Bond and Bergstrom, 2006). The comparison of characteristics of BrC in Nanjing and Beijing(Yan et al., 2015;Cheng et al., 2016;Yan et al., 2017) are supplied in this study, as mentioned in response 2). The discussions are added in Page 8, Line 8 and Page 15, Line 13-15 in revised manuscript.*

7) Line 8 on page 7: Please clarify the potential influence of dust, another important light absorber in the

atmosphere, and explain how to exclude it.

*Response: Thanks for the comment. Firstly, dust is abundant in coarse mode (Seinfeld and Pandis, 2006;Hu et al., 2013). The mass fraction of dust is low in PM$_{2.5}$ (Wang et al., 2017) except during dust storm episode. We used a PM$_{2.5}$ cutter on the inlet of Aethalometer to avoid the impact of coarse mode particles. Moreover, the mass absorption efficiency (MAE) of dust is around 1% of MAE of BC (Clark et al., 2016). Thus, the contribution of dust on PM$_{2.5}$ light absorption should be small. Figure R1 shows the time series of the concentration of Ca$^{2+}$, a tracer to dust, OC and b$_{abs\_BrC}$ measured at the SORPES station in spring (typical dust event season). It can be found that the calculated light absorption of BrC was less influenced by dust.*

[Figure]

*Figure R1. Time series of the concentration of Ca$^{2+}$, a tracer to dust, OC and b$_{abs\_BrC}$ measured at the SORPES station*

8) Eq 2: How about the influence of other light-absorbing substances, and the coating thickness/composition of black carbon?

*Response: Thanks for the comment. As mentioned in the introduction, BC, dust, and BrC have been considered as dominant contributors to aerosol absorption. In our revised manuscript, the impact of coating thickness is already considered in both SP2 data analysis and AAE$_{520-880}$. We calculate AAE$_{BC520-880}$ and AAE$_{BC370-520}$ from*

*SP2 data using core-shell Mie model. Coating thickness derived from SP2 is included in the calculation. In long-term observation, $AAE_{520-880}$ is calculated based on data from Aethalometer. Related description is shown in Page 7 to Page 10 in revised manuscript.*

9) Line 4-5 on page 8: It would be better to point out major factors which might influence the AAE values or light absorption properties of BC, before the sentence "Therefore, it is essential to firstly evaluate . . .."

***Response***: *Thanks for the comment. We have added the description in Page 9, Line 16-17 in the revised manuscript.*

*'...Previous studies have reported that the AAE of pure BC is close to 1.0, and the AAE value of 1.0 was adopted for BC by many researches (Shen et al., 2017b; Olson et al., 2015; Lack and Langridge, 2013). However, $AAE_{BC}$ can vary with BC core size, coating thickness, morphology, etc. Evidences showed that AAE of pure BC cores can be lower than 1.0 as the diameter is out of the range of Rayleigh theory, and that BC with clear shell can possibly have AAE higher than 1.0 (Bond et al., 2013; Lack and Cappa, 2010; Gyawali et al., 2009)...'*

10) Line 8 on page 10: The current display format of Figure 3 is not quite powerful for the statement of "especially in less polluted periods", as many negative values also occurred in higher polluted days, with even lower negative values.

***Response***: *Thanks for the comment. Yes, this expression is not very clear here. More precise expression should be 'when light absorption of BrC is low, more negative values occur by using the assumption of $AAE_{BC} = 1$' (shown in Figure 4 in revised manuscript). We have changed this sentence in the description of Figure 4 (shown in Page 12, Line 13-17 in revised manuscript).*

*'Also, as shown in Fig. 4, calculating $b_{abs\_BrC}$ assuming $AAE_{BC}=1$ leads to a large amount of negative values, especially when light absorption of BrC is low. While by using modified method, long-term $b_{abs\_BrC}$ can be obtained with satisfactory data validity.'*

[Figure]

*Figure 4. Comparison between the time series of daily mean $b_{abs\_BrC}$ using the modified method and using $AAE_{BC} = 1.0$. The blue diamonds represent the calculation result using $R_{AAE} = 0.65$, which is the mean value from SP2 data. Calculated $b_{abs\_BrC}$ using $AAE_{BC} = 1.0$ is plotted as the black line*

11) The findings of this study were consistent with previous studies. It is better to refer to or compared with more previous studies. For example, some studies have also pointed out that coal combustion was one of the major sources of brown carbon in north China in winter.

***Response**: Thanks for the suggestion. We compared characteristics of BrC in Nanjing and Beijing (Yan et al., 2015;Cheng et al., 2016;Yan et al., 2017). Studies conducted in Beijing have also suggested the important contribution of residential biofuel and open biomass burning on BrC in winter and summer, respectively (Yan et al., 2015;Cheng et al., 2016;Fu et al., 2013), which are consistent with our conclusions. A recent study reports that coal combustion is an important source of BrC in northern China, especially in winter (Yan et al., 2017). We have added the description of this part in Page 15, Line 13-15 in the revised manuscript.*

**Detailed replies to Referee #3**

General Comments This study provides an improved approach on deriving the brown carbon absorption from AE31 measurement, and highlights the importance in using the proper AAE to extrapolate the BC absorption from longer to shorter wavelength. If the technical part of this study could be more convincible, then it could be considered for publication. I would suggest to improve the technical part by considering the following points.

*Response: Thanks for the valuable comment. To avoid the uncertainties caused by assumed BC size distribution, we reorganized our manuscript by using BC size distribution and mixing state measured by single particle soot photometer (SP2) at the SORPES station instead of using empirical BC size distribution from previous studies. We found that our previous method is still available but we modified the method by combining SP2 data in the revision. The contribution of BrC gets larger with new parameters. Also, in the revised version we estimated the uncertainties of the calculation and updated the method and result parts. Detailed revision can be found in marked-up revised manuscript.*

**Specific comments**

1) The issue of deriving the coating content from OC/EC measurement is not just from the uncertain OM/OC ratio, but also that many of the OM may not contain BC, i.e. externally mixed, the OC/EC method would tend to largely overestimate the coating associated with BC. You could use some external results to estimate this.

*Response: Thanks for the comment. We agree that the mixing state of BC has impact on BC optical properties. We have used SP2 data to get mixing state and size of BC and revised the calculation supported by SP2 results. OCEC data is no longer used in this study. Detailed description is presented in Page 10 to Page 12 in revised manuscript.*

2) The crucial results here are from the AE31. How it has been corrected is important. It is a challenge to get the proper result from this instrument (especially at shorter wavelength). Though this may have been done in your previous publications, but it's worthy to mention here, e.g. how you get the multiple scattering and scattering correction, and how you have corrected these at different wavelength, and these may substantially affect the derived AAE because these corrections rely on the AAE as well.

*Response:*

*Thanks for the comment. For Aethalometer data correction, we implemented the correction algorithm presented by Collaud Coen et al. (2010) to correct the systematic errors of filter-based absorption measurements. The attenuation coefficient $b_{ATN}$ at each wavelength $\lambda$ is firstly calculated from*

$$b_{ATN}(\lambda) = \frac{A \cdot \Delta ATN(\lambda)}{Q \cdot \Delta t}$$

*where A and Q represent the spot size and flow rate, respectively. $\Delta ATN(\lambda)$ is the attenuation change in time step $\Delta t$. $b_{abs}$ at wavelength $\lambda$ is then obtained after correction for filter-loading effect, embedded aerosol scattering effect and multiple scattering effect by the filter fiber. The correction is performed using the Collaud Coen correction algorithm with Schmid scattering correction adopted (Collaud Coen et al., 2010;Schmid et al., 2006). The equation can be presented as*

$$b_{abs}(\lambda) = \frac{b_{ATN}}{(C_{ref} + C_{scat}) \cdot R}$$

*where R is the function for filter-loading correction calculated using the equation from Collaud Coen et al. (2010) (Eq. 13). $C_{scat}$ represents the aerosol scattering correction. To calculate $C_{scat}$, light scattering coefficients and scattering Ångström exponents measured by nephelometer are used to obtain scattering at the Aethalometer wavelengths and the constants to calculate $C_{scat}$ are taken from Arnott et al. (2005). Detailed calculation equations of R and $C_{scat}$ can be found in Collaud Coen et al. (2010). $C_{ref}$ is the multiple scattering correction factor, which is set to be 4.26 according to Collaud Coen et al. (2010). The comparison of different Aethalometer corrections (Saturno et al., 2017) shows that AAE derived by Collaud Coen correction algorithm agrees well with that from multi-wavelength reference measurement, proving the reliable AAE values from this correction. Collaud Coen correction also shows a good performance in obtaining absorption coefficients at 370 nm (Saturno et al., 2017), which is the critical wavelength in BrC segregation. Absorption coefficients are presented under Standard Temperature and Pressure (STP, i.e. 273.15 K, 1013 hPa). Detailed description is presented in Page 6 to Page 7 Line 19 in revised manuscript.*

3) The BC core size is not constant, the BC from open biomass burning or domestic solid fuel burning has a larger core than from traffic (Schwarz et al., 2008) (Liu et al., 2014). As you have already raised, the larger core will have a lower AAE. Some external data could be used to make more constrains for biomass burning BC.

*Response: Thanks for the comment. We agree that BC core size can vary under influence of different dominant sources. As suggested, to derive AAE$_{BC}$, real-time core size and coating thickness of BC measured by SP2 has been used instead of taking a fixed typical BC core size distribution. The seasonal variation of AAE$_{BC}$ is also described in revision. Detailed revision is presented in Page 10 to Page 12 in revised manuscript.*

4) The ideal approach would be combing with the SP2 measurement, such as a similar study (Liu et al., 2015), however the advantage of this study is the long-term measurement using the less-cost instrumentation, and different contributions such as open biomass burning or residential burning occurred in different months, which is interesting. If by somehow, this study could benefit by constraining some of the inputs from the existing information and doing sensitivity test.

*Response: Thanks for the comment. We have combined SP2 measurement to modify the method. Since there are lots of historical observation data conducted without SP2, it is worth to try and find a way of tracing back light absorption of BrC with satisfactory uncertainty range. At this site, firstly, we calculated AAE$_{BC}$ at short and long wavelength range based on SP2 data, and analyzed their variation ranges. We found that the ratio of AAE$_{BC370-520}$ to AAE$_{BC520-880}$ (represented as R$_{AAE}$ in revised manuscript) did not show a significant change over time, so we use the mean R$_{AAE}$ to calculate real-time AAE$_{BC370-520}$. Uncertainty and sensitivity of this method is also added in the revision. Detailed descriptions of this part is presented in Page 7 to Page 12 in revised manuscript.*

5) It is better to show the location of the experimental site and the surrounding major emissions.

*Response: Thanks for the comment. The map and emission character surrounding the site were given in our previous works, e.g. (Ding et al., 2013;Ding et al., 2016) and this is also add in Page 5 Line 23-25 in revised manuscript.*

6) Equation 4 and 5 could be merged into one, it would be useful to show the Rs-1 in time series or monthly variation.

*Response: Thanks for the suggestion. R$_{AAE}$ is used in revised manuscript instead of R$_{s-l}$ for better understanding. We have added the definition of R$_{AAE}$ in the text instead of showing as Eq. 4, which is shown in revised*

*manuscript as 'Hence a correction factor $R_{AAE}$ is defined as the ratio of $AAE_{BC370-520}$ to $AAE_{BC520-880}$ calculated from SP2 data.'. The Eq. 5 in the original manuscript is '$AAE_{BC370-520} = AAE_{520-880} \times R_{AAE}$'. Now this equation is numbered as Eq. 6 in revised manuscript Page 12 Line 5. The seasonal variation of $R_{AAE}$ is shown in box-plot Figure S2(a) in revised manuscript. During the whole observation period, $R_{AAE}$ ranges between 0.60 to 0.69 ($5^{th}$ and $95^{th}$ percentile), and the median value is 0.66, 0.69, 0.64 and 0.62 in spring, summer, autumn and winter, respectively. The detailed description of this part is presented in Page 11, Line 21 to Page 12 Line 18 in revised manuscript.*

7) Please give the refractive index you used for BC, clear coatings and brown coatings in the main plot legends.

*Response: Thanks for the suggestion. The refractive index (RI) of BC core was set to be 1.56+0.47i according to Dalzell and Sarofim (1969) and RI was 1.52+0i for clear shell (Pitchford et al., 2007). These values are also listed in revised manuscript (Page 10, Line 9-11) and Figure 1 (Page 23). The brown coating case will not be discussed in our revised manuscript, due to large uncertainties on brown coating's refractive index.*

8) Cl-/EC, as an indicator of coal combustion, needs more reference, would you be able to derive the MAE of brown carbon at different months, which will be very interesting. It looks the higher Babs/K+ ratio possibly means the Dec. BrC had a higher absorption efficiency maybe?

*Response: Thanks for the comment. After revision of the method and related results, we have also modified the source analysis part in revised manuscript. Detailed description can be found in Page 15 Line 6-21 in revised manuscript. About $MAE_{BrC}$, thanks for the suggestion and this is an interesting idea. However, mass of BrC was not measured directly at this site and we can only use OC data to convert the mass of OC to mass of BrC. Since BrC mass fraction in OM may change over time and case due to different dominant sources, especially in China where emission profile is complicated, it is hard to differentiate the change of $MAE_{BrC}$ and BrC mass fraction. In the future, we will try to calculate $MAE_{BrC}$ by using water soluble OC (WSOC) measurement and have a detail study focusing on this topic. Thanks again for your valuable suggestions.*

[revised manuscript text omitted]

---

## Referee Report (RR1)

**Comments to "*Light absorption of brown carbon in eastern China based on 3-year multi-wavelength aerosol optical property observations and an improved Absorption Ångström exponent segregation method*" by Wang et al,**

**General comments**

The manuscript has been greatly improved. Now it is suitable for publication after minor revision:

1. Page 11 Line 5-6 (in the revised manuscript): please note the peak number of $D_p$ in summer and autumn were not quite similar.

2. Page 12 Line 14-15 (in the revised manuscript): Please check "Fig. S2". Here, it should be "Figure 4".

3. Page 13 Line 6-7 (in the revised manuscript): Please check the values "from 0.6 to 1.9", which are not consistent with those in the first panel of Figure 5a.

4. Please define the abbreviations (e.g., "DJF", "MAM", "JJA", "SON") in Table 1 and Figure S4. Or, use "winter", "spring", "summer" and "autumn" as used in other Figures.

5. Please use united expressions figures, e.g., "Figure" or "Fig."

---

## Author Response (AR2)

**Wang, J., Nie, W., Cheng, Y., Shen, Y., Chi, X., Wang, J., Huang, X., Xie, Y., Sun, P., Xu, Z., Qi, X., Su, H., and Ding, A.: Light absorption of brown carbon in eastern China based on 3-year multi-wavelength aerosol optical property observations at the SORPES station and an improved Absorption Ångstrom exponent segregation method, Atmos. Chem. Phys. Discuss., https://doi.org/10.5194/acp-2018-49, in review, 2018.**

**Replies to reviewers' comments**

**Overview**

10 The authors thank the reviewers for constructive comments, they helped improving the paper. We have replied to all questions raised by the reviewer. The major changes to the paper are:

- revised Figure 2 according to suggestion of Referee 3, showing $D_p/D_c$ instead of $D_p$

- added subscript in Eq. 6 to make it clearer

- added the time series of corrected $AAE_{BC}$ at 370-520 nm and the scatter of $AAE_{BC}$ in Supplement

15 Information (SI)

Below the responses are written in cursive letters and the changes to the manuscript are in blue color or shown in red color in revised manuscript.

20 **Detailed replies to reviewers' comments**

**Detailed replies to Anonymous Referee #2**

The manuscript has been greatly improved. It is suitable for publication after minor revision:

1. Page 11 Line 5-6 (in the revised manuscript): please note the peak number of Dp in summer and autumn
25 were not quite similar.

*Response: Thank for the comment. Yes. This expression is not very accurate. We have revised Figure 2 according to the suggestion by referee 3 showing $D_p/D_c$ instead of $D_p$ and we have revised the expression of this figure accordingly.*

*'...However, the coating thickness of BC is relatively lower in winter (peak number at $D_p/D_c \sim 1.6$),*
30 *possibly due to low photochemical oxidation in this season. The coating thickness of BC is higher in spring than other seasons.'*

[Figure]

*Figure 2. (a)* *Normalized number size distributions of BC core ($D_c$) and* *(b)* *$D_p/D_c$ of BC-containing particles from SP2 measurement in four seasons*

2. Page 12 Line 14-15 (in the revised manuscript): Please check "Fig. S2". Here, it should be "Figure 4".

   **Response**: *Thank for the comment. Yes, it should be "Figure 4" here. We have corrected this in the revision.*

   *'Also, as shown in Fig. 4,…'*

3. Page 13 Line 6-7 (in the revised manuscript): Please check the values "from 0.6 to 1.9", which are not consistent with those in the first panel of Figure 5a.

   **Response**: *Thank for the comment. Here the 0.6~1.9 is the variation range of $AAE_{370-520}$ ($1^{th}$ to $99^{th}$ percentile, shown in Table 1), and the solid line in Figure 5a represents the monthly average $AAE_{370-520}$. We have added the reference "Table 1" in this sentence to make it clearer.*

   *'Moreover, $AAE_{370-520}$ shows distinct seasonal variations, which has much wider range of changing with $1^{th}$ and $99^{th}$ percentile of 0.6 and 1.9 (Table 1), than that of $AAE_{520-880}$.'*

4. Please define the abbreviations (e.g., "DJF", "MAM", "JJA", "SON") in Table 1 and Figure S4. Or, use "winter", "spring", "summer" and "autumn" as used in other Figures.

   **Response**: *Thank for the suggestion. We have revised them as "winter", "spring", "summer" and "autumn" in the revised manuscript.*

5. Please use united expressions figures, e.g., "Figure" or "Fig."

   **Response**: *Thanks for the suggestion. We have checked the expressions of figures one by one again to make sure those are correct following the "Manuscript preparation guidelines for authors" in ACP website.    shows*

*this: "The abbreviation "Fig." should be used when it appears in running text and should be followed by a number unless it comes at the beginning of a sentence, e.g.: "The results are depicted in Fig. 5. Figure 9 reveals that...".”*

*Thanks again for your valuable comments.*

**Detailed replies to Referee #3**

Thanks for the careful revision of the article, the addition of SP2 data help with the final products presented. I would recommend publication after addressing a few technical issues:

5  1. Would you need to assume the BC refractive index of (2.26,2,16) (Moteki et al., 2010) to calculate the Dp/Dc at SP2 wavelength 1064nm?

*Response: Thank for the comment. Yes. We used refractive index of BC (2.26-1.26i) at 1064 nm to calculate $D_p$. We have added the description of refractive index in the methodology in revised manuscript.*

*'...$D_p$ of BC was then estimated by using core-shell Mie model assuming BC core and shell refractive index*
10  *of 2.26-1.26i (Moteki et al., 2010) and 1.52-0i (Pitchford et al., 2007), respectively….'*

2. why not set BC refractive index according to (Bond and Bergstrom, 2006).

*Response: Thank for the comment. The BC refractive index of 1.56-0.47i (Dalzell and Sarofim, 1969) is also mentioned and used in Bond and Bergstrom (2006) (Fig. 4 and Table 4 in Bond and Bergstrom (2006)). Since*
15  *we have a similar figure (Figure S1) with the Fig.4 in Bond and Bergstrom (2006) to illustrate the theoretical variation of AAE, we adopted the same refractive index as the Fig.4 in Bond and Bergstrom (2006) in order to compare the preliminary results in the first place. Therefore, we chose to use the refractive index of 1.56-0.47i for the rest of optical calculations. We have also tested the optical calculation using other refractive index mentioned in Bond and Bergstrom (2006) and our method is still available, only the set of $R_{AAE}$ may differ (e.g.*
20  *$R_{AAE}$ and $b_{abs\_BrC}$ using 1.95-0.79i is marked in Figure R1). Impact factors of BC absorption are complicated and currently it is hard to determine these properties of BC at this site, we endeavor to get the results based on some assumptions. The main purpose of this study is to give a notification of the impact of $AAE_{BC}$ at different wavelengths on BrC optical segregation and propose an idea to determine them based on current measurements. More studies concerning the variation of refractive index, densities and other properties of BC as aging may*
25  *further modify the BC optical property researches in the future.*

[Figure]

*Figure R1. The relationship between different adopted $R_{AAE}$ value and calculated overall mean $b_{abs\_BrC}$*

3. Fig. 2a should be improved for clarification, such as using lines and markers, there seems to be a step change at 110nm? Is it dN/dlogDp? Or dM/dlogDp,I would suggest a clear unit for this. Fig. 2b, also related to the methodology section, is the Dp distribution for all populated BC, or for specified Dc range? Would be better to show the Dp/Dc distribution rather than Dp, as Dp/Dc is the one to reflect coating thickness (given winter Dc is larger).

***Response***: *Thank for the suggestion. Fig. 2a is $dN/dlogD_c$. We have modified Fig. 2 according to your suggestion and revised related description. Fig. 2a is normalized $dN/dlogD_c$ and Fig. 2b is $D_p/D_c$ distribution (shown below).*

*'...Fig. 2 shows the overall $D_c$ number size distribution and $D_p/D_c$ of BC-containing particles in four seasons. It can be found that the number size distribution of BC cores in spring, summer and autumn are in similar pattern. In winter, larger BC cores take up a higher proportion than other seasons. However, the coating thickness of BC is relatively lower in winter (peak number at $D_p/D_c \sim 1.6$), possibly due to low photochemical oxidation in this season. The coating thickness of BC is higher in spring than other seasons….'*

[Figure]

*Figure 2. (a) Normalized number size distributions of BC core ($D_c$) and (b) $D_p/D_c$ of BC-containing particles from SP2 measurement in four seasons*

4. Eq 6. seems crucial, I think it needs to be clearer how you obtained the scaling factor of AAE. There are some difference between SP2 derived AAE and Aeoth measured one (would this be partly because there is still small BrC absorption at green), could you do scatter plots for all seasons and make this number more robust.

*Response: Thank for the suggestion. We have modified Eq. 6 as 'AAE$_{BC370-520\_real-time}$ = AAE$_{520-880}$ ×R$_{AAE}$' to make a clearer understanding. AAE$_{BC370-520\_real-time}$ is the corrected real time AAE$_{BC370-520}$ derived from Aethalometer measurement and the correction factor R$_{AAE}$. The difference between SP2 derived and Aethalometer measured AAE$_{520-880}$ is possibly due to the different measurement time coverage and size range of BC by the two instruments (the lower detection limit of $D_c$ by SP2 is around 70~90 nm and mixing states of small BC-containing particles is unknown due to the detection limit of scattering signal of SP2). Figure S3 shows the seasonal mean $b_{abs}$ from Aethalometer. It is also possible that there is small BrC absorption at 520 nm, but Figure S3 indicates that the impact of BrC absorption at 520 nm should be small (since BrC have much higher AAE than BC, if BrC impact is already high at 520 nm, the slope change point would occur at longer wavelength, e.g. 590 nm). For determination of R$_{AAE}$, since we found that the ratio of AAE$_{BC370-520}$ and AAE$_{BC520-880}$ derived from SP2 didn't change much in different seasons. Therefore, we used the derived ratio R$_{AAE}$ to calculate time-dependent AAE$_{BC}$ at 370-520 nm. The scatter plot of daily mean AAE$_{BC370-520}$ and AAE$_{BC520-880}$ is shown as follow, it can be found that the slope of AAE$_{BC370-520}$/AAE$_{BC520-880}$ (R$_{AAE}$) is relatively stable and the slope is same as the overall mean value of R$_{AAE}$ (0.65).*

[Figure]

**Figure S3.** Scatter plot of seasonal mean $b_{abs}$ from Aethalometer, data points are normalized using $b_{abs}$ 880 nm

[Figure]

Figure R2. Scatter plot of daily mean $AAE_{BC370\text{-}520}$ and $AAE_{BC520\ 880}$

5.  would be helpful to give a time series of your corrected $AAE_{BC}$ which have been finally used.

*Response: Thank for the comment. Time series of corrected $AAE_{BC}$ at 370-520 nm is shown as follow and as Figure S4 in revised manuscript.*

[Figure]

**Figure S4.** Time series of corrected AAE$_{BC}$ at 370-520 nm

6. I still a bit struggle to understand the application of Babs/K+. Because K+ is mainly for open biomass burning, whereas levoglucosan is generally for more smoldering such as residential burning maybe. Fig. 5a clearly showed the periodic peak of K+ in May and Jun. so Fig. 8 is simply because the crop burning has more K+ emission (if emitting the same amount of OM)?

*Response: Thanks for the comment. Yes. The lower $b_{abs\_BrC}/K^+$ is mainly due to the intense open biomass burning in June. The main purpose of comparing $b_{abs\_BrC}$ and $K^+$ in June and December is to imply the possibility of the changing dominant source of BrC in December, so the case in June is mainly for a comparison (i.e. as a reference). Fig. 7a and Fig. 8a imply that open biomass burning is the main BrC source in June, while Fig. 8 exhibits that the slope of $b_{abs\_BrC}/K^+$ in December is significantly different from that in June, indicating the presence of other dominant BrC sources in December.*

[revised manuscript text omitted]